psychology/behaviour/cognition

generosity, COVID-19, experiments, social preferences

**Author for correspondence:**
J. Kovářík
e-mail: jaromir.kovarik@ehu.eus

# Exposure to the COVID-19 pandemic environment and generosity

P. Brañas-Garza[1], D. Jorrat[1], A. Alfonso[1], A. M. Espín[2], T. García Muñoz[3] and J. Kovářík[4,5,6]

[1]Loyola Behavioral Lab & Department of Economics, Universidad Loyola Andalucía, Sevilla, Spain
[2]Department of Anthropology, Universidad de Granada, Spain
[3]Department of Quantitative Economics, Universidad de Granada, Spain
[4]Universidad del País Vasco UPV-EHU, Bilbao, Spain
[5]CERGE-EI, Prague, Czech Republic
[6]Faculty of Economics and Faculty of Arts, University of West Bohemia, Pilsen, Czech Republic

PB-G, 0000-0001-8456-6009; DJ, 0000-0002-5038-3877;
JK, 0000-0002-0436-929X

We report data from an online experiment which allows us to study how generosity changed over a 6-day period during the initial explosive growth of the COVID-19 pandemic in Andalusia, Spain, while the country was under a strict lockdown. Participants ($n = 969$) could donate a fraction of a €100 prize to an unknown charity. Our data are particularly rich in the age distribution and we complement them with daily public information about COVID-19-related deaths, infections and hospital admissions. We find correlational evidence that donations decreased in the period under study, particularly among older individuals. Our analysis of the mechanisms behind the detected decrease in generosity suggests that expectations about others' behaviour, perceived mortality risk and (alarming) information play a key—but independent—role for behavioural adaptation. These results indicate that social behaviour is quickly adjusted in response to the pandemic environment, possibly reflecting some form of selective prosociality.

## 1. Introduction

Fairness, generosity and other manifestations of human prosociality are fundamental features of well-functioning societies, with important consequences in virtually all spheres of human socio-economic life. Their role is particularly relevant in times of hardship, when the reconstruction of economic, social and political order requires people to stick together and cooperate. Hence, how

generosity and social cohesion evolve during crises—be they economic, social or health-related—is a fundamental question.

The 2020 COVID-19 pandemic is arguably an instance of such a crisis. It represents one of the most serious global crises after World War II, affecting the life of virtually all people across the globe and generating large human, economic and social costs. From a behavioural perspective, the pandemic can be viewed as a collective action problem in which the success of the group—a region, a country or the whole of humanity—depends on individual actions. Indeed, leaders continuously appeal to individual responsibility to combat the pandemic by asking people to stay at home, avoid crowds, wear face masks, not to overconsume certain goods, etc. [1]. Most of these behaviours involve a trade-off between individual and collective interests, eventually opening room for the 'tragedy of the commons' [2,3]. Although governments have attempted to prevent some collective action issues by imposing formal restrictions and sanctions, cooperation and norm adherence—and thus the ability of societies to cope with the pandemic—have still largely relied on individual, voluntary decisions.

Indeed, Campos-Mercade *et al*. [4], Dinić & Bodroža [5] and Jordan *et al*. [6] report a positive correlation between people's prosociality and health behaviours during the pandemic (i.e. more prosocial people tend to perform more prevention behaviours). Furthermore, Jordan *et al*. [6] also show that appealing to the benefits for the well-being of others is associated with an increase in COVID-19 prevention behaviours compared with appeals to their personal benefits. That is, prosocial motives are fundamental for COVID-19 prevention (see also [7]). But, what fraction of the population lacks the intrinsic motivation to behave selflessly during a pandemic? Does the fraction adapt endogenously to the pandemic environment? Is the fraction of prosocially motivated individuals smaller or larger compared with non-pandemic times? If so, what drives any change in prosociality? The answers to these questions have important implications for the design and communication of public health campaigns since they point to a potential endogeneity issue. Policies appealing to other-regarding concerns might result ineffective and even counterproductive if they do not account for the impact of the pandemic on prosociality.

Along these lines, there is an ongoing public debate about how the pandemic environment affects people's social behaviour. Together with signs of increased solidarity (e.g. people hand-making masks for others, looking out for the most vulnerable), we have observed signs of selfishness and antisocial behaviour. Many have broken confinement, hoarded essential goods, or exhibited hostile behaviours toward 'out-group' members. Nevertheless, since we cannot effectively track the behaviour of most people and reputation is often at stake due to the public nature of many actions [8,9], one cannot make precise inferences about the impact of the pandemic on 'pure' prosociality on the basis of mere observation and such anecdotal evidence.

In this study, we report data from an online experiment which allows us to study whether human generosity toward an unknown charity changed during the early COVID-19 pandemic in Andalusia, Spain, while people were under a general lockdown. Note that, since the charity's name was unknown, our study targets one specific manifestation of prosociality, namely generosity toward an unknown cause that probably benefits strangers in need. The experiment was conducted in March 2020 in Andalusia, the most populated region in Spain; one of the countries most affected by COVID-19 at that time. The experiment started 6 days after the first total lockdown of the country and lasted for 6 days. On day 1 of our experiment, there were 17 980 confirmed cases of contagion and 982 deaths in Spain, while these figures increased to 47 610 and 3434 on day 6 (figure 1). This day Spain surpassed China (where the pandemic originated) in the number of victims.

The objective of this study is threefold. First, we study how the exposure to the COVID-19 environment, associated with home confinement and social distancing measures as well as other societal and economic consequences, correlates with generosity toward an unknown cause. Second, since behaviour might change for many reasons, we discriminate between two alternative mechanisms through which the COVID-19 pandemic and the associated phenomena might influence generosity. In particular, we consider both intrinsic social concerns and expectations about others' prosociality as potential mediating factors. By 'intrinsic social concerns', we refer to self-reported measures of social preferences that ask about the respondent's aversion to inequality, which is a key variable to explain prosocial behaviour [11].

Third, another question concerns the triggering factors behind any effect. The COVID-19 threat exerts strongly heterogeneous and clearly separable effects on the health and mortality of different population strata, imposing differing incentives for public cooperation and norm adherence across people. Age and gender are particularly relevant here: mortality rates at the time of the experiment were about 10 times higher for people over than under the age of 40 and twofold higher for males, both worldwide [1] and in

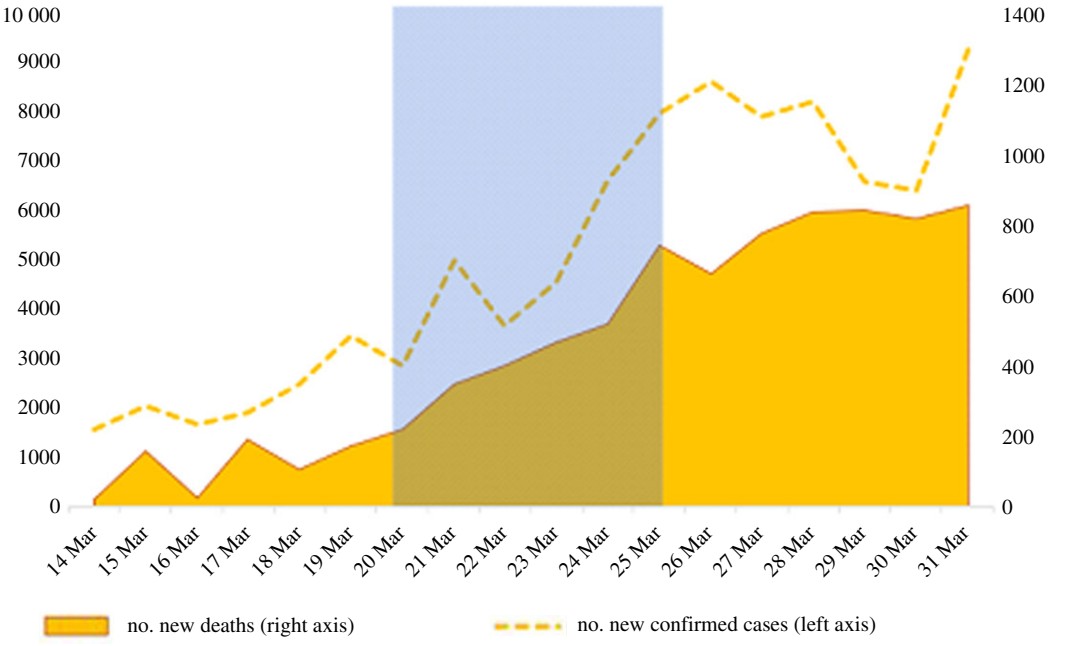

**Figure 1.** Number of new COVID-19 cases and associated deaths in Spain from the date of lockdown (14 March) to 31 March 2020. The days on which the experiment took place are highlighted (shaded area; 20–25 March). At the start of the experiment, new deaths averaged about 200/day and new cases about 3000/day. At the end of the experiment, new deaths and new cases averaged more than 700/day and 7000/day, respectively. Data source: Spanish Ministry of Health [10].

Spain [10]. During the days preceding our experiment and the days of the experiment, all the media in Spain reported the official information about the pandemic continuously and most conversations centred on it. The situation of the elderly was at the centre of the public debate. Hence, we believe that it is virtually impossible for any of our participants to be unaware of the severity and evolution of the situation, including the vulnerability of the elderly. By contrast, although the mortality rates were higher for males, the role of gender was largely absent in the media and official communications. Such information asymmetry between these factors provides an experiment-like variation in the intensity of both mortality risk (that varies across age groups and genders) and mortality cues (more salient for age than for gender). We thus ask whether generosity changes differently across people depending on the true versus perceived COVID-19-associated mortality. If both age and gender moderate the impact of the pandemic on donations, it might be argued that the *objective*, true COVID-19-associated mortality risks trigger any change in generosity, independently of the information provided in the news. However, if we find that age moderates the relationship, whereas gender does not, the data would suggest that only the *perceived* mortality risk plays a role.

With these aims, we study participants' donations to charity, their expectations about the donations of others and their self-reported social concerns (see §2). We further complement the experimental data with the official Spanish statistics regarding the number of *deaths*, *infected* people and patients at *intensive care units* (ICU) [10]. We employ the latter data as measures of the intensity of the COVID-19 threat faced by our subjects. In addition, as mentioned, age and gender are considered as potential moderators.

Although our study is exploratory, the data allow us to test a number of hypotheses based on past research for each of the three research goals discussed above. Regarding how generosity adapts to negative shocks, the arguments in the scientific literature go in both directions. The competition for scarcer resources in many such situations suggests that people might be less prosocial [2,12–14]. On the other hand, adversarial events might create a 'common enemy' effect, thus increasing cooperation and sharing [13,15–17]. The evidence supports both hypotheses (e.g. [18–23]). Importantly, these issues have been mostly analysed using aggregate data suitable to study longer-term adjustment and typically before and after the shock or only after the shock, while how prosociality adapts at the micro-level and gradually *during* a negative shock itself is understudied. Hence, previous literature does not provide us a clear hypothesis and we let the data speak.

Concerning the mediating factors, people may adapt their intrinsic social concerns endogenously to the new conditions [13,15], they might be more/less generous because they expect others to be so [24], or both. Hence, we *ex ante* expect both intrinsic social concerns and expectations to mediate the behavioural change.

Last, as for the triggering factors, past theoretical and empirical work in social psychology and human behavioural ecology explores the impact of mortality cues versus mortality risk on preferences and behaviours [25–28]. Amir *et al*. [26] is, to our knowledge, the only study within this literature analysing generosity. They report that harsher life conditions lead to higher generosity but only in reaction to mortality cues. Hence, our hypothesis is that older people—those who were continuously reminded in the news about their increased mortality risk in the study period—will exhibit stronger change in generosity, while the behavioural change will not differ across genders. In other words, the literature suggests that perceived rather than true risk will trigger the behavioural change. This moderation analysis can also shed light on the impact of other pandemic-associated factors, such as social distancing or home confinement measures. There is indeed evidence suggesting that lockdown measures may have a negative impact on prosociality [29]. Note that, since mobility restrictions affected similarly all population strata [30], no moderation effect of age or gender should be expected if lockdown measures are the main triggering factor. Alternatively, it might be argued that lockdowns impact more the youth (in terms of social behaviour and mental health, not in economic terms, which seem to be particularly relevant for older individuals; [31]) due to their higher mobility and more frequent social interactions [32,33]. This argument would suggest a moderation effect of age as well, albeit in the opposite direction: younger individuals should exhibit stronger change in generosity.

Our data provide one preliminary step toward the estimation of the extent to which people are willing to comply with the norms necessary to combat the COVID-19 pandemic and how this willingness evolves over time. Such information is critical for a correct calibration of infectious diseases models [34]. These models and the policies combating the pandemic might be miscalibrated if they do not account for the extent to which people are intrinsically motivated to act selflessly and how this motivation adapts to the pandemic environment.[1] The data generated in our study is an example of how experimental and behavioural economics can inform epidemiology and other fields in modelling disease transmission and policy makers in designing prevention policies combating the COVID-19 and other pandemics.

Naturally, the COVID-19 pandemic has already stimulated other similar studies. Five closely related studies, also analysing social behaviour within the COVID-19 pandemic context, deliver a complex message. Buso *et al*. [30] find that confined people are simultaneously more selfish in the Ultimatum Game and more cooperative in the Public Good Game, but participants who had experienced a longer lockdown (more than six weeks versus less than six weeks) are more selfish in both games. Shachat *et al*. [36] perform a longer time horizon analysis in Wuhan, China. They compare pre-epidemic data with data gathered at five points in time covering six weeks after the city's lockdown. Although their results suggest a long-term increase in prosociality along with the exposure to the pandemic, they also report a decrease (particularly, in trust) in the immediate aftermath of the Wuhan lockdown. In a survey experiment, Cappelen *et al*. [37] document that making the pandemic more salient through priming increases solidarity but also the acceptance of inequality due to luck. Adena and Harke [38] show that donations to a well-known charity (Save the Children) increase if the appeals contain a reference to COVID-19. However, all their subjects were exposed to the COVID-19 pandemic. Interestingly, Grimalda *et al*. [39], using participants from the US and Italy, and Adena & Harke [38], using participants from the UK, report conflicting evidence regarding the effects of several variables on donations to charity. Grimalda *et al*. find that personal (health) exposure increases donations, whereas Adena & Harke find a negative relationship. Moreover, environmental exposure, as measured using local COVID-19 data, also yields mixed results in these two studies. While Grimalda *et al*. find no effect of environmental exposure, Adena and Harke find a positive effect on donations. Finally, Grimalda *et al*. report an increase in generosity toward local, compared with global, COVID-19-related causes, whereas Adena and Harke find that appeals with a COVID-19 reference increase donations, but do not observe a shift toward local versus global charities.

In sum, the evidence so far indicates that social behaviour adapts following a complex pattern and that different manifestations of social preferences, selective prosociality and both short- versus long-term dynamics need to be considered separately. We believe that our data add a relevant piece of information to the debate.

---

[1]For example, the Imperial College model [35], which has guided the prevention policy in the UK while combating the COVID-19 pandemic and inspired other countries' policies, assumes among other things that 25% of older people will not comply with social distancing. This figure is assumed without any empirical basis and the parameter is held constant in their model. This may seriously misguide policy recommendations.

The remainder of the paper is organized in three sections. Section 2 describes the experimental design and procedures, §3 presents the results and the last section discusses the results and concludes.

# 2. Material and methods

## 2.1. Recruitment and sample

We invited 103 university students at an Andalusian university to participate in an online experiment and to act as recruiters. The students were encouraged and incentivized to recruit further participants from the region of Andalusia, with the objective of obtaining a richer subject pool in terms of age, non-student status and other characteristics. Gender balance and homogeneity across different ages was explicitly encouraged. Neither the participation nor recruitment were compulsory. Those who decided to participate ($n = 85$) recruited other participants from Andalusia, but also other Spanish regions and even outside Spain (see electronic supplementary material, appendix A1 for details). The initial purpose of this experiment was not to study the effect of COVID-19, as it was programmed before the pandemic surge in Spain. However, the home confinement was the reason to run the experiment online with a richer sample, rather than only with students in the laboratory.

As mentioned, the experiment focused on the region of Andalusia, but this did not prevent participation of people from outside the target region (people from other parts of Spain, $n = 191$ and from other countries, $n = 20$). Given such an initial target, and the fact that the non-Andalusian participants came from many different locations and that their numbers within locations were small and unevenly distributed, they were excluded from the main analyses.[2]

Our procedures resulted in a final sample of 969 Andalusian participants (mean age = 35.10; s.d. = 17.16) of which 55% were females. Our sample allows us to obtain small effects ($r = 0.09$) with 80% power and *alpha* = 0.05. The sample sizes for each day from 20 to 25 March were 163, 188, 139, 92, 129 and 258, respectively. Since the observations were not uniformly distributed across the 6 days of the experiment, we conservatively split the sample in half into two periods to ensure the right balance in our main analysis: 20–22 March ($n = 490$) and 23–25 March ($n = 479$). This allows us to obtain a relatively balanced sample between both 3-day periods in terms of sample size, age, education and gender (see electronic supplementary material, appendix A2 for details). Moreover, in the night of 22–23 March, the Spanish Government announced that the lockdown would be extended for (another) 15-day period, until 11 April. Therefore, the comparison between 23–25 March and 20–22 March will be our main explanatory variable (i.e. intensity of pandemic exposure).[3]

Readers might raise certain concerns regarding our experimental design, namely its online nature and the endogenous (self-)selection of the participation date in the experiment. Note that our comparison is balanced in most respects (especially, gender, age and student versus non-student participants) and we control for individual differences. Although plausible, it would be hard to claim that self-selection only operates through actual and expected donations but not through any of the variables used as controls. In addition, Snowberg & Yariv [40], using a large-scale incentivized survey, report little difference between online and laboratory behaviour and, importantly, between fast and slow participants as well as participants who endogenously participate early versus late—even those who are several times reminded to participate—in their online survey. Although we cannot rule out these concerns, they do not seem to be justified in a context like ours. Most importantly, we do not claim that we provide causal evidence and interpret our results as suggestive but highly relevant for the understanding of human prosocial behaviour and anti-pandemic policies.

All participants signed an informed consent and the data were anonymized in accordance with the Spanish Law on Personal Data Protection 3/2018. There are no participants under 16 years old.[4]

---

[2]Electronic supplementary material, table A5 in the appendix shows that the main results are robust to including the non-Andalusian subjects into our analysis.

[3]Electronic supplementary material, table A6 in the appendix provides the estimation analysis based on a linear relationship with time (i.e. days) instead of the dummy for 23–25 March. See electronic supplementary material, figure A9 for the estimated linear relationships. Even though the 6 days were not balanced in terms of the number of participants and thus this is not our preferred specification, the results are qualitatively unaffected.

[4]Those 16 and 17 years old (16 in our sample) can give their consent without asking their parents (Article 8 and Recitals 38 and 58 of the Directive 95/46/EC).

## 2.2. Experimental tasks

As is standard in economic experiments, we used monetary incentives. The instructions explained that all participants would collect points in each experimental task and these points would be converted into lottery tickets at a rate 1 point = 1 ticket. At the end of the experiment, they would participate in a lottery in which two participants would be randomly drawn and paid €100. The more lottery tickets an individual gained, the more likely she was to earn the €100. The identity and behaviour of each participant were kept anonymous to prevent reputational concerns that could affect behaviour.

The entire experimental setting consisted of several tasks (see Instructions for details[5] and electronic supplementary material, appendix A3 for the description of the experimental setting). In this paper, we focus on three behavioural measures elicited in the experiment:

(a) *Donations*. Generosity was measured using the following question: 'If you win the €100 prize, would you like to donate a fraction to an NGO?' People could choose any donation between 0% and 100%, in 10% increments. This question was incentive compatible and implemented without deceiving participants.

(b) *Expected others' donations* (not incentivized). Using the same question format, participants were asked to report their answer to the question 'How much money do you think the other participants will donate to the NGO?'. This variable also ranges from 0 to 100%, in 10% increments. In line with previous evidence (e.g. [41]), expected donations are lower than real donations (matched-pairs *t*-test, $p < 0.001$), although they are strongly correlated (Pearson's $r = 0.636$, $p < 0.001$). That is, people expect others to be less generous than themselves and those who give more expect others to give more.

(c) *Self-reported solidarity and envy*. These social preference variables measure people's self-reported aversion to advantageous inequality (often referred to as 'compassion' or 'guilt') and disadvantageous inequality [11], respectively. Using a Likert scale, we asked participants their agreement with the statement 'I do not care about how much money I have; what concerns me is that there are people who have less (more) money than I have' (proposed in [42]). The variables are labelled as SR-solidarity and SR-envy to emphasize the fact that they are based on survey *self-reports* instead of decision-making tasks [11]. As in Espín *et al*. [42], these measures predict donations (electronic supplementary material, table A2), the participants report higher SR-solidarity than SR-envy (matched-pairs *t*-test, $p < 0.001$), and the two measures are only weakly correlated (Pearson's $r = 0.117$, $p < 0.001$).

We additionally elicited certain socio-demographic variables, including gender, age, education and province of residence. Electronic supplementary material, appendices A2 and A3 provide an extensive description of the sample and the main variables of this study.

Subjects further completed the following tasks: cognitive reflection test [43,44], risk preferences [45], loss aversion [46,47], time preferences [48,49], stag hunt game [50] and big-5 personality inventory [51]. These variables are not analysed in this study (but we check the results while using them as controls).

## 2.3. Other measures

We complement our experimental data with the official Spanish statistics regarding the daily number of *deaths*, *infected* people and patients at *ICU* [10]. Since these data were released every day at 21.00 and immediately reported by virtually all Spanish media, we analyse whether the official figures from one day affect the next-day donations in our experiment. In our analysis, we interpret these figures as the measures of the perceived intensity of the pandemic threat.

## 2.4. Statistical analysis

The statistical analyses were conducted using Stata v. 15 (StataCorp). All the results reported in the tables are obtained using linear ordinary least-squares (OLS, hereafter) regressions with robust standard errors (s.e.). In the main text, we report the coefficients (coeff), standard errors and *p*-values associated with each of the hypotheses tested. The effect sizes are given using the unstandardized coefficients because

---

[5]Original instructions in Spanish and the translation to English are available at: https://repositorio.uloyola.es/handle/20.500.12412/2250.

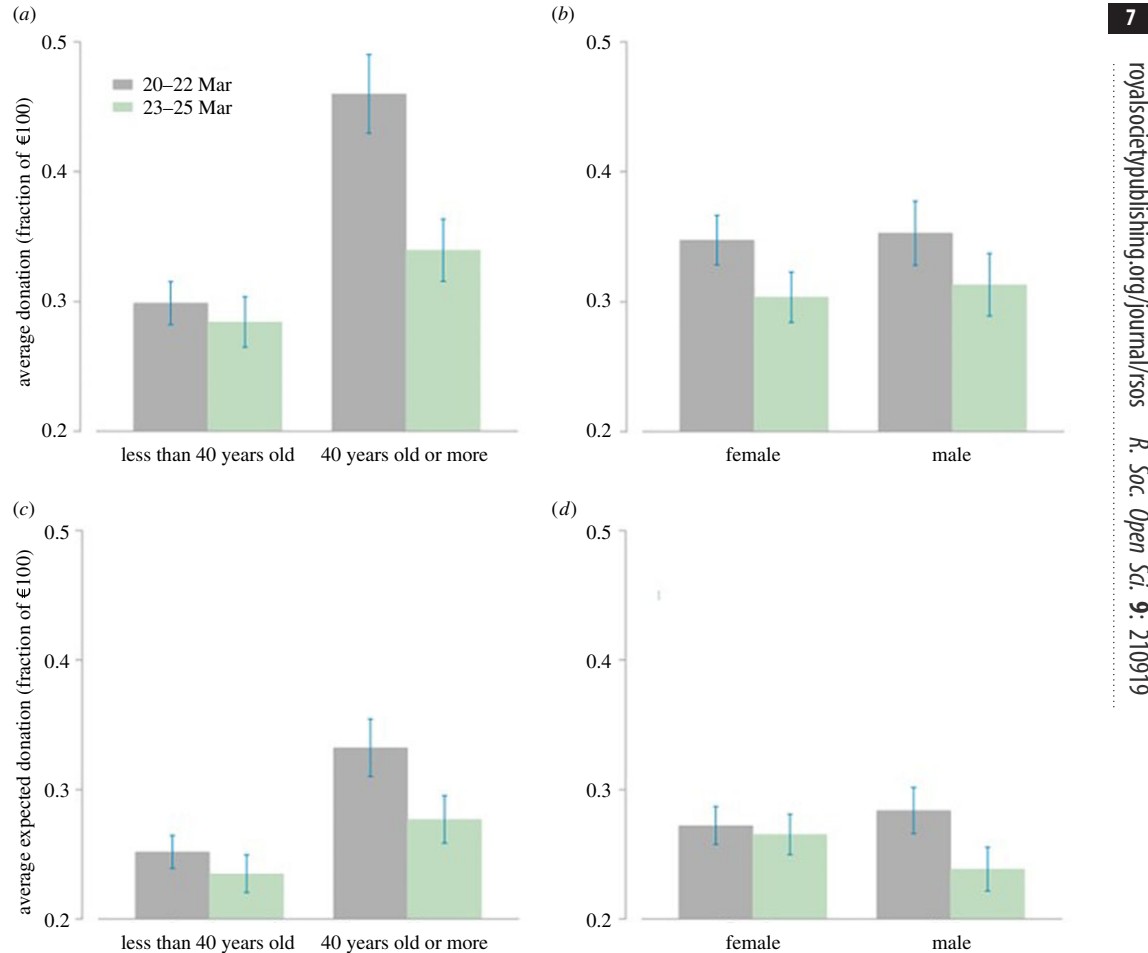

**Figure 2.** Actual and expected donations on 20–22 March versus 23–25 March. Panels (*a*) and (*b*) display average donations broken down by age groups (less than 40 years old and greater than or equal to 40 years old) and gender, respectively. Panels (*c*) and (*d*) display average expected donations broken down by age groups and gender, respectively. Error bars represent s.e.m.

the donations and expected donations are easy to interpret as a fraction of the €100 prize (i.e. each percentage point corresponds to €1). However, for the main results we also translate the unstandardized coefficients into Cohen's *d*-like effect sizes by putting them in relation to the standard deviation (s.d.) of the dependent variable.

# 3. Results

## 3.1. Giving

Figure 2 shows the donations in our experiment, disaggregated by age and gender groups (panels *a* and *b*). Panel (*a*) plots the average donations on 20–22 March versus 23–25 March for participants older or younger than 40 years old (the entire distributions of donations are shown in electronic supplementary material, figures A2 and A3 in appendix A3, where we also compare our data with Engel's [52] meta-analysis). We used 40 years old as a relevant threshold to facilitate visualization of the results in the figures; yet, the analyses were performed using age as a continuous variable. We can see that younger people are less generous overall. Nevertheless, while their donations do not seem to be affected by the COVID-19 exposure, older participants are considerably less generous in the second half of the experiment than in the first half. Panel (*b*) displays the donations disaggregated by gender (see also electronic supplementary material, figure A3 in the appendix). We do not observe any difference between males and females, and the exposure to COVID-19 does not seem to impact males and females differently. Both genders decrease their donations slightly and in similar magnitudes.

To analyse formally whether subjects' behaviour changes with the intensity of the exposure to the COVID-19 threat, model 1a in table 1 regresses participants' donations on a dummy for 23–25 March (versus 20–22 March, reflecting intensity of exposure), age (taking logs to reduce right skewness), and a male dummy. This main specification confirms that the effect of the participation day on donations is negative and statistically significant, with an estimated reduction of 6.0% in donations from 20–22 March to 23–25 March (coeff = 0.060, s.e. = 0.022, $p = 0.006$). Note that the mean donation is 32.9% with a s.d. of 33.6%. Thus, the 6% effect size corresponds to 0.18 s.d. of the dependent variable. Moreover, in line with previous evidence [52], log(age) yields a significantly positive effect on donations (coeff = 0.140, s.e. = 0.025, $p < 0.001$); gender is never significant ($p > 0.8$).

To test whether age or gender moderate the effect, model 1b in table 1 introduces the interactions *day × age* and *day × gender*. Consistent with panel (*a*) (figure 2), the *day × age* interaction is negative and significant (coeff = −0.097, s.e. = 0.049, $p = 0.048$), indicating a stronger negative effect for older participants. Using Wald tests on the model estimates, we can ask above which age the exposure decreases giving significantly. The tests suggest no effect of exposure on donations for people aged below 29 and a negative significant impact for all ages greater than or equal to 29, using $p = 0.05$. Gender, however, does not moderate the decrease in donations; the interactions *day × gender* and *day × age × gender* (model 1c) never result significant ($p > 0.4$).[6]

We thus observe that older participants donate less along with the exogenous increase in exposure to COVID-19. However, we find no differential effect by gender. In the following, we test whether subjects' expectations about others' donations and/or their self-reported social concerns explain the detected decline in donations and the differential age effect.

## 3.2. Expected giving

Expected donations (labelled as *Expectations* in table 1) report how much our participants expected others to donate. Figure 2*c,d* illustrates the effect of exposure on these expectations (see also electronic supplementary material, figure A4 in the appendix). There is an overall decline in expectations in the second half of the experiment. Notably, the effect seems to be stronger for participants above 40 (panel *c*) and males (panel *d*).

Models 2a–c reproduce models 1a–c using expected donations as the dependent variable. The estimates corroborate that higher exposure is associated with 3.4% lower expected donations in model 2a (coeff = −0.034, s.e. = 0.016, $p = 0.036$). Note that expected donations have a mean of 26.5% and a s.d. of 25.2%; thus, the 3.4% effect size corresponds to 0.13 s.d. of the dependent variable. However, although none of the interactions are significant in models 2b and 2c ($p > 0.2$), Wald tests on the estimates of model 2b again suggest the same threshold age of 29. We detect no effect of exposure for age less than 29, whereas we find a negative significant impact for all ages greater than or equal to 29 using $p = 0.05$. Regarding gender, the Wald tests on model estimates suggest no effect for females ($p > 0.4$) while we find a negative significant effect for males (coeff = −0.055, s.e. = 0.025, $p = 0.025$), as suggested by figure 2*d*.

Thus, expected donations decrease with exposure along with the actual giving. This suggests that participants might reduce their donations *because* they expect others to do so [24]. To explore this hypothesis, models 1a–c in electronic supplementary material, table A2 (appendix A4) repeat the analysis of giving to charity from table 1 introducing the expectations as an additional regressor. This model confirms the above hypothesis: the estimated effect of participation day is reduced from 6.0 to 3.2% and becomes only marginally significant once we control for expected donations (coeff = 0.032, s.e. = 0.017, $p = 0.068$; electronic supplementary material, table A2, model 1a). An equivalent structural equations model corroborates that this reduction is significant (indirect effect = −0.028, s.e. = 0.013, $p = 0.036$; electronic supplementary material, figure A7, appendix) and suggests that 47.3% of the effect of exposure on donations is mediated by participants' beliefs about others' donations.

Nevertheless, according to the Wald tests performed on the estimates from model 1b in electronic supplementary material, table A2, the negative impact of the exposure on COVID-19 remains significant at 5% for people aged over 30 (which is very close to the age of 29 found above), even if we control for the subjects' expectations. Moreover, all these results remain nearly identical if we

---

[6]The results from table 1 are robust to applying corrections for multiple hypothesis testing following Perneger [53]. The only exception is the estimated interaction between day and age in model 1b, which becomes marginally significant (the corresponding $p$-value increases from 0.048 to 0.096). Since none of the remaining results in table 1 is affected by correcting for multiple hypothesis testing, we do not report the details of the corrections here.

**Table 1.** OLS estimates: the impact of COVID-19 exposure on actual donations, expected donations and self-reported social preferences. The variable mar23–25 represents the increasing exposure to the COVID-19 threat as a dummy variable taking the value of 1 if the participation was during 23 March to 25 March and 0 if it was during 20 March to 22 March. In column 1a we regress the outcome variable donation on the participation day dummy; in column 1b we add interactions of participation day with log(age) and with gender; and in column 1c we add the three-way interaction of participation $day \times log(age) \times gender$. Columns 2a to 2c repeat the specifications of columns 1a to 1c for expected donations (beliefs). Columns 3a to 3c and 4a to 4c do similarly for self-reported solidarity and envy, respectively. All regressions include a male dummy variable and age in logs. Robust standard errors are presented in parentheses.

| | Donation | | | Expectations | | | Solidarity | | | Envy | | |
|---|---|---|---|---|---|---|---|---|---|---|---|---|
| | (1a) | (1b) | (1c) | (2a) | (2b) | (2c) | (3a) | (3b) | (3c) | (4a) | (4b) | (4c) |
| 23–25 Mar | −0.060*** | 0.273 | 0.150 | −0.034** | 0.107 | 0.077 | 0.029 | 0.212 | 0.076 | 0.001 | −0.025 | 0.071 |
| | (0.022) | (0.167) | (0.210) | (0.016) | (0.122) | (0.159) | (0.020) | (0.158) | (0.208) | (0.018) | (0.138) | (0.182) |
| Male | 0.003 | −0.000 | −0.084 | −0.010 | 0.009 | 0.033 | −0.055*** | −0.081*** | −0.137 | 0.016 | −0.003 | 0.328* |
| | (0.022) | (0.030) | (0.239) | (0.016) | (0.023) | (0.171) | (0.020) | (0.027) | (0.216) | (0.018) | (0.025) | (0.193) |
| log(age) | 0.140*** | 0.189*** | 0.179*** | 0.078*** | 0.096*** | 0.099*** | 0.099*** | 0.129*** | 0.122*** | −0.013 | −0.014 | 0.028 |
| | (0.025) | (0.035) | (0.046) | (0.018) | (0.025) | (0.032) | (0.023) | (0.032) | (0.043) | (0.020) | (0.029) | (0.041) |
| 23–25 Mar × log(age) | | −0.097** | −0.062 | | −0.036 | −0.027 | | −0.060 | −0.021 | | 0.002 | −0.026 |
| | | (0.049) | (0.063) | | (0.036) | (0.047) | | (0.046) | (0.060) | | (0.040) | (0.054) |
| 23–25 Mar × male | | 0.007 | 0.282 | | −0.037 | 0.027 | | 0.052 | 0.354 | | 0.038 | −0.193 |
| | | (0.043) | (0.342) | | (0.032) | (0.247) | | (0.040) | (0.317) | | (0.036) | (0.281) |
| male × log(age) | | | 0.025 | | | −0.007 | | | 0.017 | | | −0.098* |
| | | | (0.072) | | | (0.051) | | | (0.064) | | | (0.056) |
| 23–25 Mar × log(age) × male | | | −0.079 | | | −0.018 | | | −0.087 | | | 0.069 |
| | | | (0.100) | | | (0.072) | | | (0.092) | | | (0.081) |
| constant | −0.127 | −0.291** | −0.255* | 0.018 | −0.052 | −0.062 | 0.153* | 0.063 | 0.087 | 0.283*** | 0.295*** | 0.154 |
| | (0.082) | (0.116) | (0.150) | (0.060) | (0.084) | (0.105) | (0.078) | (0.108) | (0.145) | (0.068) | (0.097) | (0.136) |
| observations | 969 | 969 | 969 | 969 | 969 | 969 | 969 | 969 | 969 | 969 | 969 | 969 |
| adjusted $R^2$ | 0.036 | 0.038 | 0.037 | 0.019 | 0.019 | 0.017 | 0.028 | 0.029 | 0.029 | −0.002 | −0.003 | −0.001 |
| province FE | no | no | no | no | no | no | no | no | no | no | no | no |
| F-test | 11.85*** | 7.65*** | 5.74*** | 7.13*** | 4.61*** | 3.36*** | 10.10** | 6.85*** | 5.13*** | 0.347 | 0.454 | 0.855 |

$*p < 0.1$, $**p < 0.05$, $***p < 0.01$.

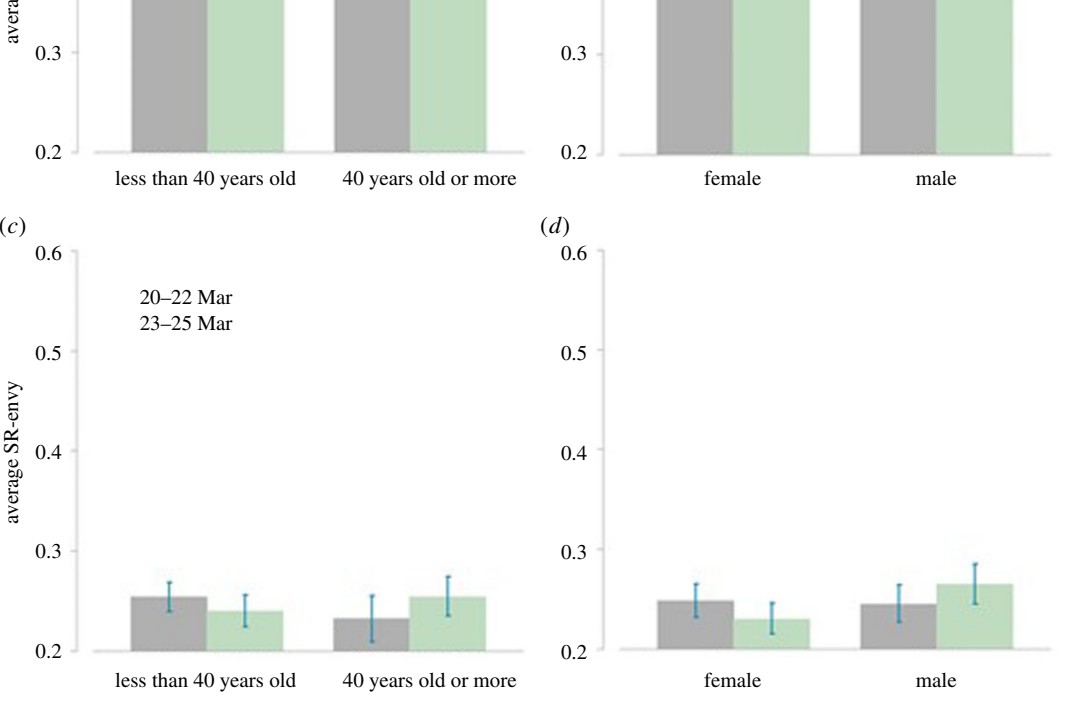

**Figure 3.** Self-reported social preferences on 20–22 March versus 23–25 March. Panels (*a*) and (*b*) display average SR-solidarity broken down for age groups (less than 40 and greater than or equal to 40) and gender, respectively. Panels (*c*) and (*d*) display average SR-envy broken down for age groups and gender, respectively. Error bars represent s.e.m.

control for risk preferences, loss aversion, time preferences, stag hunt strategy and cognitive reflection (see electronic supplementary material, appendix, table A3 for the main results controlling for these variables). Hence, expectations regarding the behaviour of others only partially mediate the age effect.

## 3.3. Self-reported inequity aversion

Last, we explore how both SR-solidarity and SR-envy vary during the confinement. Figure 3*a,b* plots the average SR-solidarity, decomposed by age and gender groups (see also electronic supplementary material, figure A5 in the appendix). The figures suggest that young participants and men declare to be somewhat less concerned about other people having less than themselves and both groups increase their solidarity slightly with the exposure (but the latter effect is insignificant in the regressions). Regarding envy, figure 3*c,d* suggests that there are no remarkable differences between females and males or age groups as well as between 20–22 March and 23–25 March (see also electronic supplementary material, figure A6 in the appendix).

Models 3a to 4c in table 1 estimate whether self-reported social preferences (solidarity and envy) also change with exposure. The analysis reveals that none of the main or interaction effects of these variables is ever significant (all $p > 0.1$). Hence, self-reported social preferences do not seem to change along with the exposure.

Finally, we use self-reported solidarity and envy as explanatory variables for giving in models 2a–c in electronic supplementary material, table A2 (appendix). As mentioned, solidarity and envy are significant predictors of donations in our experiment: those who self-report higher solidarity and lower envy donate more (coeff = 0.196, s.e. = 0.037 and coeff = −0.124, s.e. = 0.38, respectively, both $p < 0.001$). The

moderation effect of age on giving is robust to controlling for self-reported solidarity and envy though. Model 2a indicates that the effect of participation day remains negative and statistically significant (coeff = −0.066, s.e. = 0.021, $p$ = 0.002) and model 2b reveals that the negative effect of the interaction $day \times age$ remains negative and marginally significant (coeff = −0.086, s.e. = 0.049, $p$ = 0.079). The Wald test suggests no effect of exposure for age less than 24 and a negative significant impact for age greater than or equal to 24 using $p$ = 0.05 if we control for the self-reported social preferences. Most importantly, since the effect of day on donations is never reduced after controlling for self-reported preferences, these findings indicate that the detected behavioural change and the moderation effect of age *cannot* be explained by self-reported social concerns.

These are important results insofar as they talk against self-selection problems, but do not fully eliminate them. Although plausible, we find little reason to believe that participants in the first days of the experiment are more prosocial than participants in the last days due to self-selection (into participation date) but that selection only operates through actual and expected donations, not through any other variable (including self-reported social preferences).

## 3.4. Public information

The above analysis studies how donations vary between the first and second halves of the experiment. However, figure 1 shows that the number of affected and deceased people in Spain does not scale up linearly over the 6 days under study. Since these figures were continuously broadcast, they can be considered as proxies for public information regarding the intensity of exposure to COVID-19. To assess whether the data on the evolution of the pandemic provides any additional information, the models in electronic supplementary material, table A4 in the appendix substitute the time variables in our models with the official daily data regarding the new *deaths*, new *infected* people and new patients in ICU, as provided by the Spanish Ministry of Health [10] and reported by Spanish media.

Independently of the measure we use as the explanatory variable, the results are largely consistent with the above findings: subjects' donations decrease with the number of new *deaths*, *infected* and *ICU* admissions and the effect is mostly driven by older participants. The comparison between the models using day as a continuous variable (electronic supplementary material, table A6) and the models using health-related information (electronic supplementary material, table A4) reveal that employing the public data improves the fit of our regression analysis. In fact, the adjusted $R^2$ is slightly worse in the regressions in electronic supplementary material, table A6 as compared with electronic supplementary material, table A4. Since deaths, cases (infected) and ICU admissions, as mentioned, did not scale up linearly along the 6 days of the experiment, this finding would support the argument that the documented impact has more to do with the evolution of the COVID-19 pandemic threat than with the lockdown measures.

In sum, higher exposure to the pandemic environment is associated with lower donations, particularly among older participants. Gender does not moderate the effect of exposure on donations. Expectations about others' donations partially explain the effect of exposure on donations but the moderation effect of age is rather orthogonal to the mediation effect of the expectations. On the other hand, self-reported social concerns, although they predict donations, do not change along with exposure and therefore cannot explain the decrease in donations or the age interaction effect.

# 4. Discussion and conclusion

To approach how prosociality responds to 'slow disasters' (such as droughts, pandemics or a gradual erosion of global climate), we analyse how giving behaviour towards an unknown NGO changes during the early exposure to the COVID-19 pandemic in southern Spain. Although our experiment was conducted only over 6 days during the early pandemic, we observe that people decrease their generosity significantly. The decline is more pronounced for older participants, who faced higher mortality rates and they were continuously reminded in the news. By contrast, we observe no gender interaction effect, although men also had higher mortality rates (but this fact received little attention in the media).

There are several interpretations for these results. For example, people might donate less over time in our experiment because they shift their solidarity toward objectives directly linked to the COVID-19 pandemic. Note that we did not specify which charity would receive the donations. Our participants did not know whether the donated money was aimed at COVID-19-related issues. Nevertheless, our preferred interpretation—and the one more in line with existing literature—is that the COVID-19 threat may decrease generosity toward the 'out-group' or toward people not considered as part of the

'in-group', but may increase solidarity within own social circles [54–57]. Increased out-group bias during the COVID-19 pandemic has indeed been documented in Bartos *et al*. [58]. Therefore, our results might suggest some form of selective prosociality or, in other words, that people change the target of their prosocial actions in response to the pandemic environment. Our data cannot discriminate between these and other potential explanations, however; the reported evidence should be viewed as a starting point for the understanding of how and at what speed such adversarial events shape human behaviour. Similarly, we explore behavioural change in a short time window and therefore our results need to be extended to longer-term effects or adaptation processes.

Our findings have both theoretical and practical implications. Our data provide suggestive evidence that the deterioration of large-scale social capital might have contributed to the collapse of societies affected by 'slow disasters', such as droughts or pandemics, and that the process can be, paradoxically, extraordinarily fast. Such explanations have been proposed [59], but data collection under controlled environments in such cases is virtually impossible. Since most challenges that humanity currently faces are of global nature and depend on the collective response of large groups of unrelated individuals [15], if our results are corroborated by further research, they would have implications not only for social and economic post-COVID-19 recovery policies. The results would also have broader implications for the building of social resilience to the current challenges faced by humanity, such as climate change and overexploitation of natural resources, for which large-scale collective response is as important as for the COVID-19 pandemic [60].

The moderating effect of age on decreasing donations—absent for gender—points to the pivotal role of information in shaping behaviours. This finding contributes to enhance our understanding of how social and mass media, leaders and gossip influence behaviour, with implications for regulations targeting the media, the access of different age groups to certain information and fake news epidemics. Since the dynamics of donations differ from those of self-reported solidarity/envy in our data, people might be unaware to what extent these factors shape their behaviour or it might be that the context/threat modulates behaviour but not intrinsic traits (as measured by the self-reports). Future studies should explore these and other possibilities in more detail.

Needless to say, our study presents several limitations. First, our evidence is not causal. We manipulated neither the exposure to the COVID-19 threat nor the public media. Rather, we correlate the behaviour in our experiment with the participation date and the official public figures regarding the health and life impact of the pandemic. Therefore, our design cannot fully get rid of spurious relationships. The most delicate confounding factor is perhaps self-selection into the different participation days: i.e. the alternative explanation would be that individuals who participate in the last days are less generous than those participating in the first days for reasons other than exposure to the pandemic, and *the more so the older they are*. We do not find clear reasons why this should be the case, especially considering that the results do not change when we control for a battery of variables (meaning that changes in other variables, should they exist, do not explain the observed changes in actual and expected donations). Moreover, such an alternative should also explain why self-reports do *not* reflect the self-selection effect even though SR-solidarity and SR-envy predict donations similarly in the two experiment halves.[7] For these reasons, we see the self-selection alternative rather implausible, although it cannot be completely discarded. In any case, a self-selection effect, if it exists, should be extraordinarily strong (in opposition to previous relevant data showing no difference between 'slow' and 'fast' participants; [40]) to be able to more than compensate for the observed decrease in generosity among older participants. Put differently, our results virtually rule out the possibility that generosity actually *increased* among older people in the aftermath of the COVID-19 pandemic. In sum, the current findings can be seen as one (arguably relevant) piece of the puzzle that needs to be complemented with further results.

Second, our subjects were simultaneously exposed to the pandemic as well as an obligatory home confinement. We thus cannot discriminate whether the documented change in donations is due to the pandemic itself (i.e. the health-related threat), the lockdown, or both. There is evidence indicating that the lockdown can in fact be associated with less prosociality [30]. Yet, the finding of stronger effects among older participants suggests that the health-related (perceived) threat might be more influential than the lockdown measures in explaining the decrease in donations because the lockdown affected similarly all population strata, not particularly the elder (and potentially more the youth; [32,33]). Similarly, the fact that daily public information variables yield slightly better fit in our models than a

---

[7]The interaction between participation day (20–22 March versus 23–25 March) and self-reported social preferences does not significantly explain giving, either for SR-solidarity or SR-envy ($p > 0.1$).

continuous time variable (i.e. day of participation) suggests that the health-related perceived threat may influence the results more than the lockdown measures.

Third, we analyse short-run effects. As discussed above, previous research indicates that social behaviours seem to follow complex patterns that might differ between the short and long run (see more on this below).

To put our results in relation with those of previous similar studies, note that Shachat *et al.* [36] show a negative effect of pandemic exposure on prosociality in the short run but a positive effect in the long run. Buso *et al.* [30] find that longer lockdowns are associated with reduced prosociality. Our findings are aligned with these results. Relatedly, Bartos *et al.* [58] report that the pandemic has increased hostility toward foreigners. Grimalda *et al.* [39] also show that local COVID-19-related charities were favoured over global ones (but see [38], for null results in the UK). The decreased generosity detected in our study might partially explain their findings insofar as they point to some form of selective prosociality. There are other findings, however, that are harder to reconcile with ours. For example, environmental exposure (i.e. local pandemic severity) has shown no effect on donations to COVID-19-related charities in the USA and Italy [39] and a positive effect in the UK [38]. Also, personal (health) exposure to COVID-19 has been found to reduce donations to charity in the UK [38] but to *increase* donations in the USA and Italy [39].

The scarcity of data on prosociality during the early pandemic, added to the fact that these circumstances will be hardly repeated in the near future, make our correlational results particularly relevant. Future research should uncover whether our results extend to other contexts and cultures, to what extent they are causal, and progress in separating the impact of the health-related threat from that of the confinement. Nevertheless, we do believe that the reported findings deliver an important message for our understanding of human prosocial behaviour and for policies combating the pandemic.

Taken together, the existing evidence suggests that behavioural adaptation to the pandemic, particularly regarding social behaviour, does exist but also that the process is complex and multifaceted. Many aspects need to be accounted for, including the time horizon considered, expectations about others' prosociality, the specific target of people's prosocial actions and potentially cultural factors as well.

Ethics. The experiment was approved by the Ethical Committee of Loyola Andalucía University (code: 040618). All participants signed an informed consent and the data were anonymized in accordance with the Spanish Law on Personal Data Protection 3/2018.

Data accessibility. The data are available at: https://doi.org/10.5061/dryad.cnp5hqc4x.

Authors' contributions. All authors gave final approval for publication and agreed to be held accountable for the work performed therein.

Competing interests. We declare we have no competing interests.

Funding. Financial support from MINECO-FEDER (PGC2018-093506-B-I00, PID2019-106146GB-I00 and PID2019-108718GB-I00), Excelencia—Andalucía (PY18-FR-0007), the Basque government (IT1336-19), the University of Granada (B-SEJ-280-UGR20) and GAČR (17-25222S) is gratefully acknowledged. Antonio Espín acknowledges funding from the European Union's Horizon 2020 research and innovation program under the Marie Skłodowska-Curie grant agreement no. 754446 and UGR Research and Knowledge Transfer Fund – Athenea3i.

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
