## [Peer Review File · Royal Society Open Science]

Review History

RSOS-210919.R0 (Original submission)

Review form: Reviewer 1 (Pol Campos-Mercade)

Is the manuscript scientifically sound in its present form?

Yes

Are the interpretations and conclusions justified by the results?

No

Is the language acceptable?

Yes

Do you have any ethical concerns with this paper?

No

Have you any concerns about statistical analyses in this paper?

No

Recommendation?

Major revision is needed (please make suggestions in comments)

Comments to the Author(s)

See attached file (Appendix A).

Review form: Reviewer 2 (Jessica Ayers)**Is the manuscript scientifically sound in its present form?**

Yes

Are the interpretations and conclusions justified by the results?

Yes

Is the language acceptable?

No

Do you have any ethical concerns with this paper?

No

Have you any concerns about statistical analyses in this paper?

Yes

Recommendation?

Major revision is needed (please make suggestions in comments)

Comments to the Author(s)

Review of "Exposure to the COVID-19 pandemic and generosity"

Manuscript ID: RSOS-210919

In this paper, the authors describe how the COVID-19 pandemic affected generosity in Spain during the first six days of the national lockdown (March 20-25). The authors are specifically interested in determining if generosity (as indexed by donations to a charity) increased or decreased in this population during the slow burn crisis. The authors' analyses suggest 1) donations decreased over the study period, 2) access to information regarding the health impact of the pandemic (cases, deaths, number of people in the ICU) was also associated with decreased generosity, and 3) this was particularly true for older individuals in their sample.

I believe that the main points of this manuscript are interesting and assess an impactful research question (i.e., do slow-burn disasters impact the generosity of those who experience them). As the authors state, their analyses are primarily correlational but represent an essential step towards understanding the nature of generosity during pandemics and how this generosity (or lack thereof) has important implications for public health policies and national responses to the pandemic. However, I do have reservations about the publication of the manuscript in its current form. Primarily, my concerns center on the structure of the manuscript, lack of clarity in certain parts of the manuscript, and the need for a more thorough explanation in other parts of the manuscript. I feel confident that the authors would be able to address these comments with a major revision of their manuscript.

Below I provide additional details about each of my suggestions, enumerated for ease of communication during the review process:

1. The manuscript has minor spelling/ grammatical errors (e.g., use of different tenses in the same paragraph, plural vs. singular nouns and verbs). While this is not a significant issue (as these are issues I also commonly run into in my writing), I found myself distracted by these instances and unable to focus on the authors' argument. I feel that the authors would be able to address these instances with another read-through of their manuscript.

2. In the abstract, it would help to have a little more precision and clarity. For example, why not use the exact number of participants instead of a rounded approximation? Why not separate the results to state their importance clearly? Why not list that solidarity and empathy are the "social preferences" that are studied? As is, it feels like the reader needs to have read the manuscript in its entirety to fully understand and appreciate the work the authors present in the manuscript.

3. I'm not sure if it was in the submission process or conversion to a PDF for reviewers, but the version of the manuscript I received had not indentations for paragraphs. This was highly distracting, as there were multiple places in the manuscript where it appeared as if paragraphs went on for pages and switched topics in numerous sites. It would help (if only for the sake of reviewing) to make sure that new paragraphs are indicated with indentations or lines between paragraphs.

4. I found the introduction to be a little simplistic in its presentation of information. While the studies and topics that the authors mention are interesting, it feels as if the authors assume that their readers have read the same primary sources and do not need a detailed explanation of the studies, variables, and topics presented as the rationale for their research. Examples of this include: "Indeed, CamposMercade et al. (2021) report a positive correlation between people's prosociality and health behaviors during the pandemic.", "Not surprisingly, scholars across the behavioral sciences call to emphasize the impact of one's own actions on the well-being of others while designing and communicating public health campaigns (Van Bavel et al., 2020) and many campaigns in fact do so." While the authors say these references support their rationale, it is not clear to me as a reader how they support it. What variables, outcomes, and conclusions from these articles lead them to study generosity? I needed to refer to these citations and briefly skim the articles to understand their rationale fully. Since this isn't practical for other readers, it would help if more information about these citations were given in the introduction to fully set up the rationale for the study.

5. The authors try to claim that their results speak to selfish vs. prosocial responses in their sample. But, this is only one explanation for their results. It is also possible that the participants are employing selective generosity that their results are not able to capture - for example, it is possible that participants would have been more likely to be generous with romantic partners (Lukaszewski and Roney 2010), friends (Krems et al. 2021), or those in their community who need their help or with whom the participant pools their risk (Aktipis 2016; Cronk et al. 2019; Cronk et al. 2019). None of these alternative explanations discount the work and conclusions drawn by the authors, but to state that their work addressed participants who lack "intrinsic motivation" to behave selflessly during this time is not accurate.

6. More explanation is needed in the introduction to set up the later use of solidarity and envy as social preferences that may influence generosity in this sample. From my reading, there is only one paragraph (on page 4) that focuses on "observed" signs of solidarity and anti-social behaviors, but there is a lack of citations to relevant literature (such as Mesurado et al. 2021; Dinić and Bodroža 2020; Buso et al. 2020), so the reader is asked to take the authors' word on this. More explanation, and nuance of how this may manifest, is needed.

7. The paper describes the study's objectives (3), but does not provide concrete and testable hypotheses that they have designed the study to test. If the study is exploratory (as I suspect it is),

that is perfectly fine, but the authors should 1) state that the study is exploratory and 2) detail the potential outcomes for their objectives and what the outcomes would mean for generosity during the pandemic (i.e., that people were more or less generous during the lockdown).

8. The authors state: “We thus ask whether generosity changes differently across people depending on the true vs. perceived Covid-19-associated mortality” But, having read through the manuscript, it is not clear how they do this. Making the operationalization of these concepts and the subsequent hypothesis and meaning transparent would help readers feel guided through this part of the experiment.

9. The structure of the paper would be significantly improved if the authors implemented a more traditional introduction-method-results-discussion and conclusion format. The interpretation and implications of the results are currently presented in tandem with previous research in the introduction and, to put it plainly, this format makes it hard to follow the logic that motivated the current study. By interpreting the results in the introduction (before the reader has had a chance to read and evaluate the results for themselves; pages 5 - 8), the reader just has to trust that they would interpret the results similarly. For example, statements like: “The decreased generosity detected in our study might partially explain these results.”, “While our results are in line with the latter evidence, all these findings indicate that social behavior adapts following a complex pattern and that both short-term and long-term dynamics need to be considered separately.”, “Last, as mentioned above, Campos-Mercade et al. (2021) report correlations between social preferences and health behaviors, highlighting the importance of our findings for the effectiveness of frequent public recommendations and policies appealing to human other-regarding concerns during the pandemic.” imply that the reader already knows the results and can assess the accuracy of these claims for themselves, something that is not possible in an introduction before results are shown. The discussion of the results (pg 5 - 8) is more important for a discussion section, though many of the ideas and descriptions need more explanation for readers to assess them easily.

10. Throughout the manuscript, the authors refer to one variable as “contagion” and “cases.” It would be best to refer to the same variable by a single term, and “cases” is probably the most accurate (since the variable reports the number of cases in Spain for that day).

11. The methods section mentions that participants were incentivized to further recruit participants - how was this done?

12. As far as the method section describes, this study was co-opted for this purpose (i.e., it was originally planned for another purpose). If this is not the case, the writing needs to be clearer to describe how the authors made the study in response to the announced lockdowns. Currently, the impression is that students were already participating, and the purpose of the study changed once the lockdown was announced (which calls into question whether a sensitivity analysis or a priori power analysis is needed - It is currently also unclear which the authors present here).

13. When the authors talk about their balanced sample, it is a little unclear if they mean balance between the March 20-22 and March 23-25 groups or balanced in terms of generalizability to the overall Spanish population.

14. It is unclear what the experimental earnings are and how they were converted into lottery tickets (e.g., is 1 euro = 1 lottery ticket?; how much did people earn on average?).

15. There are many potential DVs mentioned in the text that are not fully described in the results - where are the results of the cognitive reflection, risk preferences, loss aversion, time preferences,

stag hunt, and Big - 5 variables? The manuscript says all other variables, but that doesn't give enough information.

16. It would be very helpful to discuss the overall regression analysis results (and type of regression used) in the text on page 12 instead of directing the readers to the table for the information.

17. The kinds of coefficients reported needs to be more explicit. Regression analysis gives both standardized and unstandardized coefficients, and there is nowhere in the text that specifies what coefficient the authors are interpreting & comparing across models (though, hopefully, it is the standardized coefficients).

18. Results, coefficients, standard errors, and p values jump between percentages and decimals. Please pick the format you intend to use for these variables and make it consistent throughout the manuscript.

19. Please give the coefficients/ effect sizes for the effects described - giving p-values does not give the reader enough information to assess the size of the effect beyond saying that the effect exists (which is not surprising given the large sample). I noticed this starting on page 14.

20. It is very odd to report results that are coming from another preprint (page 14, footnote 12), and I am not sure this inclusion buys the authors much in terms of explanations as it is not fully clear why this other analysis from the same data but in another paper is necessary. My reading of this part was that there wasn't a massive influence of day x age when controlling for day x expectations, but it is important enough to be in another paper, so it seems like the authors are misleading the reader as to somethings else regarding this analysis that could be important. Since I do not think this is the authors' intention, it might be best to add this as a footnote or completely omit it from the current manuscript.

21. It would be helpful to know the kind of coefficients for tables 1 & 2, and to specify that the numbers in parentheses mean SE. It is also unclear what "Province FE" refers to.

22. There were a lot of tests in this manuscript but no mention of corrections for multiple tests. It would be helpful for the authors to mention which of these effects survive correcting for multiple tests.

23. It is unclear why the authors begin referring to solidarity and envy as "SR-solidarity" and "SR-envy" as all of their data is self-report (even the donations). There are no objective measures in this study simply because the experiment was conducted online while people were stuck at home. In addition, the authors have some rationale in the introduction as to why they thought solidarity and envy would be important, but it is still not clear why these measures and not others were used.

24. The authors state " The moderation effect of age on giving is robust to controlling for self-reported solidarity and envy though." but it is not clear what analysis the authors are referring to as Table 1 suggests envy and solidarity were in separate models and outcome variables.

25. I believe that the authors mean that exposure to public information (e.g., deaths, cases, ICU patients) causes donations to decrease. Still, the wording "donations scale down" implies that the donations decrease in a scaleable form as more access/ exposure to information is included in the model. As that is not how table 2 presents the data, more clarity is needed here.

26. It is unclear what the authors mean with “Shachat et al. (2020) perform a longer time horizon analysis in Wuhan, China and find long-term increases in prosociality after a decline in the immediate aftermath of the city lockdown. Our findings are consistent with such a pattern and, together, suggest that the behavioral adaptation process may follow complex dynamics.” pg 20. The “behavioral adaptation process” aspect is not clear - do you mean changes in prosociality?

27. This section is also not clear (pg. 20) “The results would also have broader implications for the building of social resilience to future disasters, for which large-scale collective response is as important as for the Covid-19 pandemic. Such challenges are numerous (Boyd et al., 2018).”

28. By this point in the discussion, the effects of age are referred to as moderating effects - but this language was not used in the results section. It would be helpful to call the age effects similar things across all sections of the paper (for consistency from the reader’s perspective, I had to go back and make sure the effect was a moderation effect instead of something else).

29. Toplak et al. is missing from the reference section.

30. I’m not sure if I missed it in my download, but I did not have access to the Appendix. I also was not able to access the link to the anonymous data file from the link provided.

While my critiques of the manuscript seem numerous, I would like to reassure the authors that this is only the case because I firmly believe there are many more positive aspects of the manuscript than things I can critique. My main concern primarily deals with the framing of the manuscript, which is why I have suggested a major revision.

As per my policy, I sign all of my reviews. The author should feel free to contact me if they have any questions or points of clarification regarding my review. I truly enjoyed this manuscript and applaud the authors for their excellent work.

Jessica D. Ayers
jdayers@asu.edu

Decision letter (RSOS-210919.R0)

Dear Dr Kovarik

The Editors assigned to your paper RSOS-210919 "Exposure to the Covid-19 pandemic and generosity" have now received comments from reviewers and would like you to revise the paper in accordance with the reviewer comments and any comments from the Editors. Please note this decision does not guarantee eventual acceptance.

We invite you to respond to the comments supplied below and revise your manuscript. Below the referees’ and Editors’ comments (where applicable) we provide additional requirements. Final acceptance of your manuscript is dependent on these requirements being met. We provide guidance below to help you prepare your revision.

Please submit your revised manuscript and required files (see below) no later than 21 days from today's (ie 29-Sep-2021) date. Note: the ScholarOne system will 'lock' if submission of the revision is attempted 21 or more days after the deadline. If you do not think you will be able to meet this deadline please contact the editorial office immediately.

on behalf of Dr Simone Schnall (Associate Editor) and Essi Viding (Subject Editor)
openscience@royalsociety.org

Associate Editor Comments to Author (Dr Simone Schnall):

Associate Editor: 1

Comments to the Author:

Dear Dr. Kovarik and Colleagues,

Thank you for submitting your manuscript to our journal and please excuse the delay in communicating with you. It was exceptionally difficult to find reviewers for your paper, no doubt as a result of the challenging pandemic working conditions, and because Covid-19 has sparked considerable research interest, resulting in many important findings.

I am therefore especially grateful to the two Reviewers, both experts on the topic, who provided thorough evaluations. They are both enthusiastic about your research, and I share their positive views. The comments range from somewhat major (e.g., relevant literature and theoretical rationale) to more minor (e.g., typos and formatting) and I suggest you attend to them as much as you can. You will find the relevant points in the reviews below, which are self-explanatory so I will not repeat them here.

Please note that the journal's default revision time is rather short, but it would be completely acceptable if you wanted to take longer to revise the paper, to make it as strong as possible. I look forward to receiving the revision in due time.

Sincerely,
Simone Schnall
Associate Editor

Reviewer comments to Author:

Reviewer: 1

Comments to the Author(s)

See attached file.

Reviewer: 2

Comments to the Author(s)

Review of "Exposure to the COVID-19 pandemic and generosity"

Manuscript ID: RSOS-210919

In this paper, the authors describe how the COVID-19 pandemic affected generosity in Spain during the first six days of the national lockdown (March 20-25). The authors are specifically interested in determining if generosity (as indexed by donations to a charity) increased or decreased in this population during the slow burn crisis. The authors' analyses suggest 1) donations decreased over the study period, 2) access to information regarding the health impact of the pandemic (cases, deaths, number of people in the ICU) was also associated with decreased generosity, and 3) this was particularly true for older individuals in their sample.

I believe that the main points of this manuscript are interesting and assess an impactful research question (i.e., do slow-burn disasters impact the generosity of those who experience them). As the authors state, their analyses are primarily correlational but represent an essential step towards understanding the nature of generosity during pandemics and how this generosity (or lack thereof) has important implications for public health policies and national responses to the pandemic. However, I do have reservations about the publication of the manuscript in its current form. Primarily, my concerns center on the structure of the manuscript, lack of clarity in certain parts of the manuscript, and the need for a more thorough explanation in other parts of the manuscript. I feel confident that the authors would be able to address these comments with a major revision of their manuscript.

Below I provide additional details about each of my suggestions, enumerated for ease of communication during the review process:

1. The manuscript has minor spelling/ grammatical errors (e.g., use of different tenses in the same paragraph, plural vs. singular nouns and verbs). While this is not a significant issue (as these are issues I also commonly run into in my writing), I found myself distracted by these instances and unable to focus on the authors' argument. I feel that the authors would be able to address these instances with another read-through of their manuscript.
2. In the abstract, it would help to have a little more precision and clarity. For example, why not use the exact number of participants instead of a rounded approximation? Why not separate the results to state their importance clearly? Why not list that solidarity and empathy are the "social preferences" that are studied? As is, it feels like the reader needs to have read the manuscript in its entirety to fully understand and appreciate the work the authors present in the manuscript.
3. I'm not sure if it was in the submission process or conversion to a PDF for reviewers, but the version of the manuscript I received had not indentations for paragraphs. This was highly distracting, as there were multiple places in the manuscript where it appeared as if paragraphs went on for pages and switched topics in numerous sites. It would help (if only for the sake of reviewing) to make sure that new paragraphs are indicated with indentations or lines between paragraphs.

4. I found the introduction to be a little simplistic in its presentation of information. While the studies and topics that the authors mention are interesting, it feels as if the authors assume that their readers have read the same primary sources and do not need a detailed explanation of the studies, variables, and topics presented as the rationale for their research. Examples of this include: "Indeed, CamposMercade et al. (2021) report a positive correlation between people's prosociality and health behaviors during the pandemic.", "Not surprisingly, scholars across the behavioral sciences call to emphasize the impact of one's own actions on the well-being of others while designing and communicating public health campaigns (Van Bavel et al., 2020) and many campaigns in fact do so." While the authors say these references support their rationale, it is not clear to me as a reader how they support it. What variables, outcomes, and conclusions from these articles lead them to study generosity? I needed to refer to these citations and briefly skim the articles to understand their rationale fully. Since this isn't practical for other readers, it would help if more information about these citations were given in the introduction to fully set up the rationale for the study.

5. The authors try to claim that their results speak to selfish vs. prosocial responses in their sample. But, this is only one explanation for their results. It is also possible that the participants are employing selective generosity that their results are not able to capture - for example, it is possible that participants would have been more likely to be generous with romantic partners (Lukaszewski and Roney 2010), friends (Krems et al. 2021), or those in their community who need their help or with whom the participant pools their risk (Aktipis 2016; Cronk et al. 2019; Cronk et al. 2019). None of these alternative explanations discount the work and conclusions drawn by the authors, but to state that their work addressed participants who lack "intrinsic motivation" to behave selflessly during this time is not accurate.

6. More explanation is needed in the introduction to set up the later use of solidarity and envy as social preferences that may influence generosity in this sample. From my reading, there is only one paragraph (on page 4) that focuses on "observed" signs of solidarity and anti-social behaviors, but there is a lack of citations to relevant literature (such as Mesurado et al. 2021; Dinić and Bodroža 2020; Buso et al. 2020), so the reader is asked to take the authors' word on this. More explanation, and nuance of how this may manifest, is needed.

7. The paper describes the study's objectives (3), but does not provide concrete and testable hypotheses that they have designed the study to test. If the study is exploratory (as I suspect it is), that is perfectly fine, but the authors should 1) state that the study is exploratory and 2) detail the potential outcomes for their objectives and what the outcomes would mean for generosity during the pandemic (i.e., that people were more or less generous during the lockdown).

8. The authors state: "We thus ask whether generosity changes differently across people depending on the true vs. perceived Covid-19-associated mortality" But, having read through the manuscript, it is not clear how they do this. Making the operationalization of these concepts and the subsequent hypothesis and meaning transparent would help readers feel guided through this part of the experiment.

9. The structure of the paper would be significantly improved if the authors implemented a more traditional introduction-method-results-discussion and conclusion format. The interpretation and implications of the results are currently presented in tandem with previous research in the introduction and, to put it plainly, this format makes it hard to follow the logic that motivated the current study. By interpreting the results in the introduction (before the reader has had a chance to read and evaluate the results for themselves; pages 5 - 8), the reader just has to trust that they would interpret the results similarly. For example, statements like: "The decreased generosity detected in our study might partially explain these results.", "While our results are in line with

the latter evidence, all these findings indicate that social behavior adapts following a complex pattern and that both short-term and long-term dynamics need to be considered separately.”, “Last, as mentioned above, Campos-Mercade et al. (2021) report correlations between social preferences and health behaviors, highlighting the importance of our findings for the effectiveness of frequent public recommendations and policies appealing to human other-regarding concerns during the pandemic.” imply that the reader already knows the results and can assess the accuracy of these claims for themselves, something that is not possible in an introduction before results are shown. The discussion of the results (pg 5 - 8) is more important for a discussion section, though many of the ideas and descriptions need more explanation for readers to assess them easily.

10. Throughout the manuscript, the authors refer to one variable as “contagion” and “cases.” It would be best to refer to the same variable by a single term, and “cases” is probably the most accurate (since the variable reports the number of cases in Spain for that day).

11. The methods section mentions that participants were incentivized to further recruit participants - how was this done?

12. As far as the method section describes, this study was co-opted for this purpose (i.e., it was originally planned for another purpose). If this is not the case, the writing needs to be clearer to describe how the authors made the study in response to the announced lockdowns. Currently, the impression is that students were already participating, and the purpose of the study changed once the lockdown was announced (which calls into question whether a sensitivity analysis or a priori power analysis is needed - It is currently also unclear which the authors present here).

13. When the authors talk about their balanced sample, it is a little unclear if they mean balance between the March 20-22 and March 23-25 groups or balanced in terms of generalizability to the overall Spanish population.

14. It is unclear what the experimental earnings are and how they were converted into lottery tickets (e.g., is 1 euro = 1 lottery ticket?; how much did people earn on average?).

15. There are many potential DVs mentioned in the text that are not fully described in the results - where are the results of the cognitive reflection, risk preferences, loss aversion, time preferences, stag hunt, and Big - 5 variables? The manuscript says all other variables, but that doesn't give enough information.

16. It would be very helpful to discuss the overall regression analysis results (and type of regression used) in the text on page 12 instead of directing the readers to the table for the information.

17. The kinds of coefficients reported needs to be more explicit. Regression analysis gives both standardized and unstandardized coefficients, and there is nowhere in the text that specifies what coefficient the authors are interpreting & comparing across models (though, hopefully, it is the standardized coefficients).

18. Results, coefficients, standard errors, and p values jump between percentages and decimals. Please pick the format you intend to use for these variables and make it consistent throughout the manuscript.

19. Please give the coefficients/ effect sizes for the effects described - giving p-values does not give the reader enough information to assess the size of the effect beyond saying that the effect exists (which is not surprising given the large sample). I noticed this starting on page 14.

20. It is very odd to report results that are coming from another preprint (page 14, footnote 12), and I am not sure this inclusion buys the authors much in terms of explanations as it is not fully clear why this other analysis from the same data but in another paper is necessary. My reading of this part was that there wasn't a massive influence of day x age when controlling for day x expectations, but it is important enough to be in another paper, so it seems like the authors are misleading the reader as to somethings else regarding this analysis that could be important. Since I do not think this is the authors' intention, it might be best to add this as a footnote or completely omit it from the current manuscript.

21. It would be helpful to know the kind of coefficients for tables 1 & 2, and to specify that the numbers in parentheses mean SE. It is also unclear what "Province FE" refers to.

22. There were a lot of tests in this manuscript but no mention of corrections for multiple tests. It would be helpful for the authors to mention which of these effects survive correcting for multiple tests.

23. It is unclear why the authors begin referring to solidarity and envy as "SR-solidarity" and "SR-envy" as all of their data is self-report (even the donations). There are no objective measures in this study simply because the experiment was conducted online while people were stuck at home. In addition, the authors have some rationale in the introduction as to why they thought solidarity and envy would be important, but it is still not clear why these measures and not others were used.

24. The authors state "The moderation effect of age on giving is robust to controlling for self-reported solidarity and envy though." but it is not clear what analysis the authors are referring to as Table 1 suggests envy and solidarity were in separate models and outcome variables.

25. I believe that the authors mean that exposure to public information (e.g., deaths, cases, ICU patients) causes donations to decrease. Still, the wording "donations scale down" implies that the donations decrease in a scaleable form as more access/ exposure to information is included in the model. As that is not how table 2 presents the data, more clarity is needed here.

26. It is unclear what the authors mean with "Shachat et al. (2020) perform a longer time horizon analysis in Wuhan, China and find long-term increases in prosociality after a decline in the immediate aftermath of the city lockdown. Our findings are consistent with such a pattern and, together, suggest that the behavioral adaptation process may follow complex dynamics." pg 20. The "behavioral adaptation process" aspect is not clear - do you mean changes in prosociality?

27. This section is also not clear (pg. 20) "The results would also have broader implications for the building of social resilience to future disasters, for which large-scale collective response is as important as for the Covid-19 pandemic. Such challenges are numerous (Boyd et al., 2018)."

28. By this point in the discussion, the effects of age are referred to as moderating effects - but this language was not used in the results section. It would be helpful to call the age effects similar things across all sections of the paper (for consistency from the reader's perspective, I had to go back and make sure the effect was a moderation effect instead of something else).

29. Toplak et al. is missing from the reference section.

30. I'm not sure if I missed it in my download, but I did not have access to the Appendix. I also was not able to access the link to the anonymous data file from the link provided.

While my critiques of the manuscript seem numerous, I would like to reassure the authors that this is only the case because I firmly believe there are many more positive aspects of the manuscript than things I can critique. My main concern primarily deals with the framing of the manuscript, which is why I have suggested a major revision.

As per my policy, I sign all of my reviews. The author should feel free to contact me if they have any questions or points of clarification regarding my review. I truly enjoyed this manuscript and applaud the authors for their excellent work.

Jessica D. Ayers
jdayers@asu.edu

===PREPARING YOUR MANUSCRIPT===

===PREPARING YOUR REVISION IN SCHOLARONE===

Author's Response to Decision Letter for (RSOS-210919.R0)

See Appendix B.

Decision letter (RSOS-210919.R1)

Dear Dr Kovarik,

It is a pleasure to accept your manuscript entitled "Exposure to the Covid-19 pandemic environment and generosity" in its current form for publication in Royal Society Open Science. The comments of the reviewer(s) who reviewed your manuscript are included at the foot of this letter.

Please ensure that you send to the editorial office an editable version of your accepted manuscript, and individual files for each figure and table included in your manuscript. You can send these in a zip folder if more convenient. Failure to provide these files may delay the processing of your proof.

COVID-19 rapid publication process:

We are taking steps to expedite the publication of research relevant to the pandemic. If you wish, you can opt to have your paper published as soon as it is ready, rather than waiting for it to be published the scheduled Wednesday.

This means your paper will not be included in the weekly media round-up which the Society sends to journalists ahead of publication. However, it will still appear in the COVID-19 Publishing Collection which journalists will be directed to each week (<https://royalsocietypublishing.org/topic/special-collections/novel-coronavirus-outbreak>).

If you wish to have your paper considered for immediate publication, or to discuss further, please notify openscience_proofs@royalsociety.org and press@royalsociety.org when you respond to this email.

Please see the Royal Society Publishing guidance on how you may share your accepted author manuscript at <https://royalsociety.org/journals/ethics-policies/media-embargo/>. After

publication, some additional ways to effectively promote your article can also be found here <https://royalsociety.org/blog/2020/07/promoting-your-latest-paper-and-tracking-your-results/>.

on behalf of Dr Simone Schnall (Associate Editor) and Essi Viding (Subject Editor)
openscience@royalsociety.org

Associate Editor Comments to Author (Dr Simone Schnall):

Dear Kovarik,

I now had an opportunity to consider your revised manuscript. You did an exceptionally thorough job in addressing the reviewers' comments, for which you should be commended. It is therefore my pleasure to accept your paper for publication.

Congratulations on such a strong piece of work, and thanks for choosing Royal Society Open Science as an outlet to share your findings.

Sincerely,
Simone Schnall
Associate Editor, RSOS

Appendix A

Referee report: Exposure to the Covid-19 pandemic and generosity

I enjoyed a lot reading the paper! I think that the authors do a fantastic job at motivating and writing it. I also believe that understanding the effects of the pandemic is very relevant, and I appreciate the effort that the authors have put in trying to understand these effects on social preferences. I believe that this paper could make a great publication for this journal. However, I think that the paper would benefit from addressing a few issues.

Here are my main comments:

- 1) - Is the effect that the authors find a result of being exposed to the pandemic or to the lockdown? Drawing from both personal experience and previous literature, I suspect it is more likely to be the latter: When the experiment started, people were already in lockdown and very likely felt that they were exposed to the pandemic. After one week of being in lockdown, I am not convinced that seeing on TV that there are 500 deaths, rather than the 400 deaths that appeared yesterday, makes people feel “more exposed to the pandemic”. Rather, people may be more sensitive to the fact that they have been closed at home, without the freedom to do what they like to do (meeting up with friends, playing football, or going hiking) for a long time. The authors may want to take a look at Layard et al. (2020), Andersson et al. (2021), and Giuntella et al. (2021) who discuss that the psychological costs of lockdowns can be substantial.
 - Embracing the lockdown explanation would yield a much more consistent story with the rest of the literature. Most papers find that exposure to the pandemic fosters social preferences (Shachat et al. 2020; Cappelen et al. 2020). There are also two recent papers that I have reviewed that find that the pandemic increases cooperation to fight climate change (unfortunately, there are no working papers for those papers, but they should be out in a few weeks/months). However, it seems that lockdowns may damage social preferences in the short run (Buso et al. 2020; Shachat et al. 2020). Hence, the effect that the authors find would be very much in line with the literature if they instead interpret them as being due to having stayed 2 weeks (rather than 1) under lockdown and without freedom of movement.
 - If the authors agree that, given their experience and the rest of the literature, it is likely that they are capturing lockdown effects, I would encourage them to slightly change the focus of the paper in that direction. If they do not agree, then I believe that the paper should discuss in detail this alternative explanation and why the authors think it is not a likely one.
 - I would like to highlight that, regardless of the route that the authors decide to take, in my view the scientific value of the paper would remain intact.
- 2) The text often implies that the fact that deaths/cases/ICU cases correlate with generosity indicates that the effect on generosity comes from exposure to this information. However, if generosity decreases over time, and if these measures increase over time, the correlation is very likely to be there even if these measures have nothing to do with

generosity. So, personally, I do not think that this analysis brings much to our understanding of the mechanisms. I would tone down these discussions and potentially send the analysis to the appendix.

- 3) Given that all Spain went into lockdown, it seems weird to exclude 191 participants because they were not Andalusian. I understand the point that these may be unevenly distributed across time, but the econometric analysis should be able to deal with this by including region fixed effects.
- 4) I appreciate that the authors split the sample into two halves, one for the first and one for the second period. I think that this helps interpreting the results and generating easily interpretable figures. However, the effects of pandemic/lockdown exposure should in theory increase daily (rather than once mid-week), so including the “time” variable linearly would in my view make a more correct specification. I would provide these results, together with a figure showing the average donation made every day, either in the main text or in the appendix.

Minor comments:

- At the end of page 12, why do the authors divide age in +/- 29 years old? Is there some reason behind this cutoff? The wording could also be improved (“we accept a negative significant impact at 5%”). Similarly, in page 16 the authors use a cutoff of +/- 24 years old. But in the rest of the paper the authors use +/- 40. Since the analysis was not pre-registered, I believe it is important to stick to one definition.
- Proposal (feel free to disregard): I would give the correct number of participants in the abstract, rather than an approximation.
- Proposal (feel free to disregard): The last sentence in the abstract does not read great. I am also not sure whether the information contained there is relevant enough.
- Proposal (feel free to disregard): I would mention the shortcomings in the conclusion rather than in the introduction.
- Proposal (feel free to disregard): The title could be more informative by replacing “and” for a verb. For example, “Exposure to the Covid-19 pandemic reduces generosity” or “Exposure to lockdowns reduces generosity”
-

Typos:

- Second paragraph, second line: change “crisis” for “crises”
- 4th page, second line: “in case of covid-19” for “in the case of covid-19”
- 8th page, third line: “but this not prevented” for “but this did not prevent”

REFERENCES:

Andersson, O., Campos-Mercade, P., Carlsson, F., Schneider, F., & Wengström, E. (2021). The Individual Welfare Costs of Stay at Home Policies. Forthcoming at *Scandinavian Journal of Economics*.

Buso, I. M., De Caprariis, S., Di Cagno, D., Ferrari, L., Larocca, V., Marazzi, F., ... & Spadoni, L. (2020). The effects of COVID-19 lockdown on fairness and cooperation: Evidence from a lablike experiment. *Economics Letters*, 196, 109577.

Cappelen, A. W., Falch, R., Sørensen, E. Ø., & Tungodden, B. (2021). Solidarity and fairness in times of crisis. *Journal of Economic Behavior & Organization*, 186, 1-11.

Giuntella, O., Hyde, K., Saccardo, S., & Sadoff, S. (2021). Lifestyle and mental health disruptions during COVID-19. *Proceedings of the National Academy of Sciences*, 118(9).

Layard, R., Clark, A., De Neve, J. E., Krekel, C., Fancourt, D., Hey, N., & O'Donnell, G. (2020). When to release the lockdown? A wellbeing framework for analysing costs and benefits.

Shachat, J., Walker, M. J., & Wei, L. (2020). The impact of the Covid-19 pandemic on economic behaviours and preferences: Experimental evidence from Wuhan.

Appendix B

Reply to Reviewer 1:

Dear Reviewer, first of all, we would like to thank you for your positive assessment of our manuscript and for all the feedback you have provided. Your comments were very constructive and helped us to improve considerably the manuscript.

In what follows, we reply to your comments one by one. For a better readability of our reply, we preserve your comments in this document in *italics* and reply on them right below the corresponding points (starting with the word "REPLY:"). Please, note that, apart from the modifications due to your comments, there is quite a large number of changes made on request of Reviewer 2.

1) Is the effect that the authors find a result of being exposed to the pandemic or to the lockdown? Drawing from both personal experience and previous literature, I suspect it is more likely to be the latter: When the experiment started, people were already in lockdown and very likely felt that they were exposed to the pandemic. After one week of being in lockdown, I am not convinced that seeing on TV that there are 500 deaths, rather than the 400 deaths that appeared yesterday, makes people feel "more exposed to the pandemic". Rather, people may be more sensitive to the fact that they have been closed at home, without the freedom to do what they like to do (meeting up with friends, playing football, or going hiking) for a long time. The authors may want to take a look at Layard et al. (2020), Andersson et al. (2021), and Giuntella et al. (2021) who discuss that the psychological costs of lockdowns can be substantial. Embracing the lockdown explanation would yield a much more consistent story with the rest of the literature. Most papers find that exposure to the pandemic fosters social preferences (Shachat et al. 2020; Cappelen et al. 2020). There are also two recent papers that I have reviewed that find that the pandemic increases cooperation to fight climate change (unfortunately, there are no working papers for those papers, but they should be out in a few weeks/months). However, it seems that lockdowns may damage social preferences in the short run (Buso et al. 2020; Shachat et al. 2020). Hence, the effect that the authors find would be very much in line with the literature if they instead interpret them as being due to having stayed 2 weeks (rather than 1) under lockdown and without freedom of movement. If the authors agree that, given their experience and the rest of the literature, it is likely that they are capturing lockdown effects, I would encourage them to slightly change the focus of the paper in that direction. If they do not agree, then I believe that the paper should discuss in detail this alternative explanation and why the authors think it is not a likely one. - I would like to highlight that, regardless of the route that the authors decide to take, in my view the scientific value of the paper would remain intact.

REPLY: We do agree. The argument that the documented effects may be influenced by the home confinement is obviously valid. However, our data do not allow us to conclude whether the observed effects are due to the pandemic threat, the home confinement, or both. Although our analysis of moderators suggests that the health-related (perceived) threat does influence the change in generosity (note that lockdown restrictions affected everyone similarly but the perceived health-related threat focused on the elder), we cannot discard that the lockdown itself also has an impact. It might be that the lockdown affects especially older individuals, but we see this explanation somewhat less reasonable for the age moderation effect than the threat-related explanation. In fact, the literature points to a stronger negative effect of lockdowns among the youth (Czeisler et al., 2020; Giuntella et al., 2021), which would suggest a moderation effect of age in the opposite direction as we find. In addition, the health-related information data provide a slightly better fit to the data than a linear time regressor (see reply to comments #2 and #4), which also talks against the interpretation of the lockdown as the main driver. That said, we opted for an intermediate discourse, in which we admit this possibility and acknowledge that we cannot fully separate the two. This approach can be visible in the new title (we included the word "environment" to make clear that we refer to the pandemic situation, with all its numerous associated factors), the abstract, and throughout the whole Introduction section. In addition, in Discussion and Conclusion, we follow your suggestion and discuss this issue in detail and in relation to the literature.

However, we now review the existing literature in more detail both in the Introduction and the Discussion and Conclusion sections and conclude that the evidence regarding the effect of the Covid-19 pandemic is rather mixed even within studies. For example, Shahat et al. observe an initial decrease and later increase

in generosity. Cappelen et al. make the pandemic more salient through priming, showing that it increases solidarity but also the acceptance of inequality due to luck. However, all their subjects are exposed to the Covid-19 pandemic. Hence, we prefer to be more cautious in the manuscript without making any general inferences regarding this point (but still recognizing the possibility that you raise in this comment).

We hope that you like the way we addressed this comment.

2) The text often implies that the fact that deaths/cases/ICU cases correlate with generosity indicates that the effect on generosity comes from exposure to this information. However, if generosity decreases over time, and if these measures increase over time, the correlation is very likely to be there even if these measures have nothing to do with generosity. So, personally, I do not think that this analysis brings much to our understanding of the mechanisms. I would tone down these discussions and potentially send the analysis to the appendix.

REPLY: The comparison of the models with time as a continuous variable (Table A6; built to address your comment #4) and the public data (Table 2 in the original version) suggests that the employment of the public figures slightly improves the model performance. This suggests that public information might be a fundamental factor, but the results are by no means conclusive in this respect. Therefore, following your suggestion, we reduced considerably the corresponding discussion in the Results section and moved most of it—including the original Table 2—into the Appendix A4 (the old Table 2 became Table A4 in the new version). In addition, we discuss this issue in several parts of the current version of the paper.

3) Given that all Spain went into lockdown, it seems weird to exclude 191 participants because they were not Andalusian. I understand the point that these may be unevenly distributed across time, but the econometric analysis should be able to deal with this by including region fixed effects.

REPLY: Since the experiment from the very beginning targeted Andalusian subjects, we had little control over the participation outside of the region. We do not know the reasons why people from outside the target region participated but these might be spurious. As a result, the non-Andalusian subjects are unevenly distributed both across time and space. In addition, the non-Andalusian participants differ from the Andalusian ones in some aspects: e.g. they are more educated (difference=-0.59, $p<0.001$) and younger (difference= -2.94, $p<0.001$). Note also that, given the low numbers in some of the regions (see below), the region fixed effects may be blurring part of the effect of the remaining variables (in particular, exposure). We therefore maintain the models with Andalusian subjects only as the benchmark analysis.

Region	Freq.	Percent	Cum.
Andalucia	969	83.61	83.62
Aragon	2	0.17	83.79
Asturias	5	0.43	84.22
Cantabria	12	1.03	85.26
Castilla Leon	2	0.17	85.43
Castilla la Mancha	23	1.98	87.41
Catalunya	16	1.38	88.79
Comun. Valenciana	41	3.53	92.33
Comun. de Madrid	58	5	97.33
Extremadura	10	0.86	98.19
Galicia	4	0.29	98.45
Navarra	7	0.6	99.05
Pais Vasco	8	0.69	99.74
Islas	3	0.26	100
Total	1,160	100	

This notwithstanding, following your suggestion, we include all the subjects and use region fixed effects in regressions reported in Table A5 in the Appendix (mentioned additionally in a footnote in the main text), where you can see that our results do not change qualitatively.

4) I appreciate that the authors split the sample into two halves, one for the first and one for the second period. I think that this helps interpreting the results and generating easily interpretable figures. However, the effects of pandemic/lockdown exposure should in theory increase daily (rather than once mid-week), so including the “time” variable linearly would in my view make a more correct specification. I would provide these results, together with a figure showing the average donation made every day, either in the main text or in the appendix.

REPLY: Similarly to your comment #3, since the observations are unequally distributed across the six days under study (see Table A1), we opted for being more conservative and employ the time dummy. This analysis is kept in Table 1.

However, upon your request, Table A6 in the Appendix now reproduces our regressions using the proposed linear relationship as a robustness check. One again, the results are qualitatively unaffected with one exception: under this model specification, the dayxage interaction is still negative but no longer significant. However, the Wald test employed on the model estimates still detects a negative significant relationship for people aged 30 or more while the effect for people aged 29 or less is non-significant, using $p=0.05$ as a threshold (in this respect, see also our reply to your first Minor Comment below). In addition, as requested, Figure A9 plots the average donations day by day, and the estimated linear relationship, disaggregated by age.

Minor comments:

- At the end of page 12, why do the authors divide age in +/- 29 years old? Is there some reason behind this cutoff? The wording could also be improved (“we accept a negative significant impact at 5%”). Similarly, in page 16 the authors use a cutoff of +/- 24 years old. But in the rest of the paper the authors use +/- 40. Since the analysis was not pre-registered, I believe it is important to stick to one definition.

REPLY: We realize that we might have not explained ourselves well: the main figures fix one age exogenously (40 years) and look at the below/above 40 difference for visual clarity.

However, for the regression analysis we use age as a continuous variable and since our study is explanatory (and this is clearly expressed in several parts of the paper), as is standard when the moderator is continuous, we also ask: “Above which threshold age is the effect significant in the data?” This “endogenous” threshold age, obtained through Wald tests on the coefficients evaluated at the different ages, is then reported in the text. This explains why the number differs depending on the model or type of analysis. To make this distinction clear for future readerships, we explain this better in the manuscript. Please see the Results section. Thank you for pointing our attention to the confusion this might generate.

Moreover, note that such an endogenously obtained age is very similar across models with the same dependent variable: it is 29 or 30 in all the models in which generosity is the dependent variable. The age of 24 shows up only in the regressions that control for self-reported solidarity/envy. We therefore find it useful to maintain this exercise in the analysis.

- Proposal (feel free to disregard): I would give the correct number of participants in the abstract, rather than an approximation.

- Proposal (feel free to disregard): The last sentence in the abstract does not read great. I am also not sure whether the information contained there is relevant enough.

- Proposal (feel free to disregard): I would mention the shortcomings in the conclusion rather than in the introduction.

- *Proposal (feel free to disregard): The title could be more informative by replacing “and” for a verb. For example, “Exposure to the Covid-19 pandemic reduces generosity” or “Exposure to lockdowns reduces generosity”*

REPLY: All changed as suggested. In the abstract, we report the exact number of subjects and removed the last sentence. In addition, we moved the discussion of shortcomings into the Discussion and Conclusion section and (due to your comment #1) changed the title to “Exposure to the Covid-19 pandemic **environment** and generosity.” We included the word “environment” to reflect that with the pandemic we refer to all the conditions of the pandemic (including the home confinement but also other phenomena). We also prefer not to include “reduces” in the title because the discussion of the literature clearly argues in favor of a more complex pattern than just “reduces” (see our reply to your comment #1 and the Discussion and Conclusion section in the manuscript). Given this, we prefer not to give such a strong message in the title. We hope you like it.

Typos:

- *Second paragraph, second line: change “crisis” for “crises”*
- *4th page, second line: “in case of covid-19” for “in the case of covid-19”*
- *8th page, third line: “but this not prevented” for “but this did not prevent”*

REPLY: All fixed. Thank you.

Reply to Reviewer 2:

Dear Jessica, thank you very much for your positive assessment of our work and for such a detailed reading of and comments on our paper. There is no doubt that addressing your comments has considerably improved our paper. Thank you very much for that.

In what follows, we reply to all your comments one by one. For a clearer exposition, we leave your comments in this reply in *italics* and reply to each of them right below the corresponding comment (each time starting with the word “REPLY:”). Please, note that, apart from the modifications due to your comments, there are numerous other changes made on request on Reviewer 1.

1. The manuscript has minor spelling/ grammatical errors (e.g., use of different tenses in the same paragraph, plural vs. singular nouns and verbs). While this is not a significant issue (as these are issues I also commonly run into in my writing), I found myself distracted by these instances and unable to focus on the authors’ argument. I feel that the authors would be able to address these instances with another read-through of their manuscript.

REPLY: A native speaker proofread the manuscript. We hope that there are no remaining errors in the text.

2. In the abstract, it would help to have a little more precision and clarity. For example, why not use the exact number of participants instead of a rounded approximation? Why not separate the results to state their importance clearly? Why not list that solidarity and empathy are the “social preferences” that are studied? As is, it feels like the reader needs to have read the manuscript in its entirety to fully understand and appreciate the work the authors present in the manuscript.

REPLY: We incorporated all your suggestions (as well as the suggestions of Reviewer 1). We report the exact number of subjects and (in response to Reviewer 1) we removed the mentioned sentence referring to social preferences and the result regarding the public information. As a consequence of the latter, the results—we hope—are separated more clearly: we present in one phrase the main result (donations decrease with time, particularly among older participants) and in another phase we summarize all the suggested mechanisms behind the documented effect. We hope that you like the new version of the abstract and consider it self-explanatory.

3. I’m not sure if it was in the submission process or conversion to a PDF for reviewers, but the version of the manuscript I received had not indentations for paragraphs. This was highly distracting, as there were multiple places in the manuscript where it appeared as if paragraphs went on for pages and switched topics in numerous sites. It would help (if only for the sake of reviewing) to make sure that new paragraphs are indicated with indentations or lines between paragraphs.

REPLY: We introduced a space after each paragraph.

4. I found the introduction to be a little simplistic in its presentation of information. While the studies and topics that the authors mention are interesting, it feels as if the authors assume that their readers have read the same primary sources and do not need a detailed explanation of the studies, variables, and topics presented as the rationale for their research. Examples of this include: “Indeed, CamposMercade et al. (2021) report a positive correlation between people’s prosociality and health behaviors during the pandemic.”, “Not surprisingly, scholars across the behavioral sciences call to emphasize the impact of one’s own actions on the well-being of others while designing and communicating public health campaigns (Van Bavel et al., 2020) and many campaigns in fact do so.” While the authors say these references support their rationale, it is not clear to me as a reader how they support it. What variables, outcomes, and conclusions from these articles lead them to study generosity? I needed to refer to these citations and briefly skim the articles to understand their rationale fully. Since this isn’t practical for other readers, it would help if more information about these citations were given in the introduction to fully set up the rationale for the study.

REPLY: We apologize for any confusion we might have caused. Due to your comments and on request of Reviewer 1, the Introduction (and also the Discussion and Conclusion section) have been modified to a large extent. We have now clarified the reasons why the cited studies are relevant to ours and many studies are now only cited in Discussion and Conclusions.

5. The authors try to claim that their results speak to selfish vs. prosocial responses in their sample. But, this is only one explanation for their results. It is also possible that the participants are employing selective generosity that their results are not able to capture - for example, it is possible that participants would have been more likely to be generous with romantic partners (Lukaszewski and Roney 2010), friends (Krems et al. 2021), or those in their community who need their help or with whom the participant pools their risk (Aktipis 2016; Cronk et al. 2019; Cronk et al. 2019). None of these alternative explanations discount the work and conclusions drawn by the authors, but to state that their work addressed participants who lack “intrinsic motivation” to behave selflessly during this time is not accurate.

REPLY: You are obviously right. We do discuss different explanations of our results in the Discussion and Conclusion section. We clearly state that there are several explanations behind our results. In fact, the explanation that we stress seems to be close to what you call “selective generosity” (labelled as *in-group bias*). Now, we mention the selective prosociality hypothesis several times in the paper (including the abstract).

6. More explanation is needed in the introduction to set up the later use of solidarity and envy as social preferences that may influence generosity in this sample. From my reading, there is only one paragraph (on page 4) that focuses on “observed” signs of solidarity and anti-social behaviors, but there is a lack of citations to relevant literature (such as Mesurado et al. 2021; Dinić and Bodroža 2020; Buso et al. 2020), so the reader is asked to take the authors’ word on this. More explanation, and nuance of how this may manifest, is needed.

REPLY: We clarified that the two inequality aversion components, “solidarity” (or compassion/guilt) and “envy”, are particularly relevant to understand social behavior in general. Please note that the inequality aversion model of Fehr and Schmidt (1999) is perhaps the most influential approach to social preferences in the economics literature. Thus, the choice of the inequality aversion self-reports has nothing to do with the “observed signs of solidarity and anti-social behaviors”. Instead, they were chosen as relevant measures of intrinsic social concern. We also cited the suggested papers.

7. The paper describes the study’s objectives (3), but does not provide concrete and testable hypotheses that they have designed the study to test. If the study is exploratory (as I suspect it is), that is perfectly fine, but the authors should 1) state that the study is exploratory and 2) detail the potential outcomes for their objectives and what the outcomes would mean for generosity during the pandemic (i.e., that people were more or less generous during the lockdown).

REPLY: We restructured the Introduction considerably. It now states clearly—right after the objectives and the description of what we do—(i) that the study is exploratory and (ii) outlines the hypotheses of our study based on the previous literature.

8. The authors state: “We thus ask whether generosity changes differently across people depending on the true vs. perceived Covid-19-associated mortality” But, having read through the manuscript, it is not clear how they do this. Making the operationalization of these concepts and the subsequent hypothesis and meaning transparent would help readers feel guided through this part of the experiment.

REPLY: We apologize for not being clear on this point. This refers to the potential moderators of the effect, i.e., age and gender. After the mentioned sentence, we added the following to clarify: “If both age and gender moderate the impact of the pandemic on donations, it might be argued that the *objective*, true Covid-19-associated mortality risks trigger any change in generosity, independently of the information provided in the news. However, if we find that age moderates the relationship whereas gender does not, the data would suggest that only the *perceived* mortality risk plays a role.” Also, the text explaining the hypotheses regarding

this point contains an additional explanation of this issue. We hope that these modifications clarify what we mean by the sentence.

9. The structure of the paper would be significantly improved if the authors implemented a more traditional introduction-method-results-discussion and conclusion format. The interpretation and implications of the results are currently presented in tandem with previous research in the introduction and, to put it plainly, this format makes it hard to follow the logic that motivated the current study. By interpreting the results in the introduction (before the reader has had a chance to read and evaluate the results for themselves; pages 5 - 8), the reader just has to trust that they would interpret the results similarly. For example, statements like: "The decreased generosity detected in our study might partially explain these results.", "While our results are in line with the latter evidence, all these findings indicate that social behavior adapts following a complex pattern and that both short-term and long-term dynamics need to be considered separately.", "Last, as mentioned above, Campos-Mercade et al. (2021) report correlations between social preferences and health behaviors, highlighting the importance of our findings for the effectiveness of frequent public recommendations and policies appealing to human other-regarding concerns during the pandemic." imply that the reader already knows the results and can assess the accuracy of these claims for themselves, something that is not possible in an introduction before results are shown. The discussion of the results (pg 5 - 8) is more important for a discussion section, though many of the ideas and descriptions need more explanation for readers to assess them easily.

REPLY: Following your suggestions and those of Reviewer 1, we restructured some parts of the manuscript considerably. In particular, all the cited statements now appear in the Discussion and Conclusion section. The introduction now only motivates, briefly mentions what we do, and provides hypotheses (see your comment #7 above). Hope that you like this structure better.

10. Throughout the manuscript, the authors refer to one variable as "contagion" and "cases." It would be best to refer to the same variable by a single term, and "cases" is probably the most accurate (since the variable reports the number of cases in Spain for that day).

REPLY: The word "contagion" only appeared once as "the cases of contagion" in Figure 1. We removed the word contagion.

11. The methods section mentions that participants were incentivized to further recruit participants - how was this done?

REPLY: This information was provided in the Appendix (Appendix A1) of the manuscript, which—for some reason—you did not receive (see you comment #30). We elaborated that information a little bit more to make it clearer. In brief, the experiment was part of the module "Game Theory for Social Sciences". One part of the module was devoted to experiments. In this part, students participated in the experiment. They were given instructions for how to recruit other participants considering the composition of the sample with respect to age and gender and how they would be paid. We weighted the three criteria (number of recruited participants, age and gender composition) and the recruiting individual with highest score received 100 points while the remaining recruiters were graded proportionally with respect to the best one. Appendix A1 describes the procedures in more detail.

12. As far as the method section describes, this study was co-opted for this purpose (i.e., it was originally planned for another purpose). If this is not the case, the writing needs to be clearer to describe how the authors made the study in response to the announced lockdowns. Currently, the impression is that students were already participating, and the purpose of the study changed once the lockdown was announced (which calls into question whether a sensitivity analysis or a priori power analysis is needed - It is currently also unclear which the authors present here).

REPLY: We apologize that we were not clear enough because this is important: we clearly reveal in both the original and the current versions of the ms. that "[t]he main purpose of this experiment was not to study the effect of Covid-19, as it was programmed before the pandemic surge in Spain. However, the home

confinement was the reason to run the experiment online...” That is, the study was indeed not designed to explore generosity during the lockdown; we have never claimed the contrary. The experiment was planned long before its implementation (as part of a course; see point #11 above). However, due to confinement we were forced to run it online. The lockdown was completely orthogonal to the study and we took the opportunity to analyze the consequences of the pandemic situation. Both Section II and Appendix A1 describe all these details.

However, the students were **not** participating in the experiment before the confinement. Figure 1 shows that the data collection started 6 days after the national lock-down. If we collected any data before that, we would definitely use them in the study. Unfortunately, we did not.

13. When the authors talk about their balanced sample, it is a little unclear if they mean balance between the March 20-22 and March 23-25 groups or balanced in terms of generalizability to the overall Spanish population.

REPLY: Whenever we mention the balanced sample, we refer to the balance between *March 20-22 and March 23-25*. We never argue that our sample is representative of the Spanish population as it clearly is not. Since the words “balanced” and “representative” have different meaning in statistics so that interchanging them might confuse the readers, we made sure that this does not happen in the new version of the manuscript. In fact, the word “representative” does not appear there.

Apart from this, Section II contains the word “balanced” several times. In one place, we write: “Gender balance and homogeneity across different ages was explicitly encouraged.” In this case, we believe that there is no room for confusion as the word “balanced” is directly linked to gender/age and “encouraged” clearly suggests that we tried to get a gender and age balance (and we deliver descriptive statistics regarding these variables in the Appendix). In the same section, we additionally state: “This allows us to obtain a relatively balanced sample between both three-day periods in terms of sample size, age, education and gender”. This a correct use of the word as it clearly refers to similar distributions of the mentioned variable across the two time intervals. Last, later in the same page, we relatedly say that the comparison is balanced, which again, we hope, leaves no room for confusion.

This notwithstanding, if you still have any issue with any of these statements, please, let us know and we will change it following your suggestion.

14. It is unclear what the experimental earnings are and how they were converted into lottery tickets (e.g., is 1 euro = 1 lottery ticket?; how much did people earn on average?).

REPLY: Once again, Appendix A1 describes these procedures in detail (so you were not able to see this information without having the appendix). In brief, participants were informed that in each experimental task they may earn experimental points. At the end of the whole experiment, the total amount of earned points were converted into lottery tickets (1 point = 1 ticket). The more tickets a subject collected, the more likely she was to be selected in the lottery. Once finished, we draw 2 ticket numbers and the 2 holders of these 2 tickets received 100 Euro each. Hence, the average earning is 200/number of participants.

15. There are many potential DVs mentioned in the text that are not fully described in the results - where are the results of the cognitive reflection, risk preferences, loss aversion, time preferences, stag hunt, and Big - 5 variables? The manuscript says all other variables, but that doesn't give enough information.

REPLY: This study focuses on generosity and variables related to generosity. We do not include in the analysis all the other variables unrelated to generosity. This is now clearly stated in Section II: “These variables are not analyzed in detail in this study.”

Since we do not use these variables in our study, we do not explain them to save space, but we feel obliged to recognize that they formed part of the experimental elicitation while explaining the design. The manuscript is already long enough for the standards of the Royal Society Open Science and describing them in detail

would take several pages so we just refer to the literature for interested readers. To the best of our knowledge, this is a common practice. If you insist on including the whole description, please, let us know, but keep in mind that it will make the article considerably larger.

16. It would be very helpful to discuss the overall regression analysis results (and type of regression used) in the text on page 12 instead of directing the readers to the table for the information.

REPLY: We have added a subsection to Section II labeled as “Statistical analysis” in which we describe the statistical strategy followed and have reported all the coefficients and SEs in the main text to facilitate reading. The subsection also defines all abbreviations used in the paper (OLS, SE, SD, coeff.,...).

17. The kinds of coefficients reported needs to be more explicit. Regression analysis gives both standardized and unstandardized coefficients, and there is nowhere in the text that specifies what coefficient the authors are interpreting & comparing across models (though, hopefully, it is the standardized coefficients).

REPLY: As stated in the “Statistical analysis” subsection in Section II (see your comment #16), “[t]he effect sizes are given using the unstandardized coefficients because the donations and expected donations are easy to interpret as a fraction of the €100 prize (i.e., each percentage point corresponds to €1). However, for the main results we also translate the unstandardized coefficients into Cohen’s d-like effect sizes by putting them in relation to the standard deviation of the dependent variable.”

18. Results, coefficients, standard errors, and p values jump between percentages and decimals. Please pick the format you intend to use for these variables and make it consistent throughout the manuscript.

REPLY: We have used a consistent reporting style throughout the new version of the ms.

19. Please give the coefficients/ effect sizes for the effects described - giving p-values does not give the reader enough information to assess the size of the effect beyond saying that the effect exists (which is not surprising given the large sample). I noticed this starting on page 14.

REPLY: Fixed. Please see our reply to comments #16 and #17.

20. It is very odd to report results that are coming from another preprint (page 14, footnote 12), and I am not sure this inclusion buys the authors much in terms of explanations as it is not fully clear why this other analysis from the same data but in another paper is necessary. My reading of this part was that there wasn't a massive influence of day x age when controlling for day x expectations, but it is important enough to be in another paper, so it seems like the authors are misleading the reader as to somethings else regarding this analysis that could be important. Since I do not think this is the authors' intention, it might be best to add this as a footnote or completely omit it from the current manuscript.

REPLY: As far as we know, not reporting the details of all the robustness checks is a common practice, especially if they corroborate the reported results. Once again, by not including the whole analysis we save on a lot of space (another table and its interpretations only to tell that the benchmark models lead to the same result). Hence, we left the discussion of robustness analysis but, as suggested, removed the reference to the more exhaustive analyses in the Psyarxiv version of the manuscript.

21. It would be helpful to know the kind of coefficients for tables 1 & 2, and to specify that the numbers in parentheses mean SE. It is also unclear what “Province FE” refers to.

REPLY: Fixed. Please see our reply to comments #16 and #17. The meaning of the parentheses is now given in the tables notes. The Province FE meant “province fixed effects” but, since in the last version of the ms. none of the regressions in the main text controls for province for the sake of conciseness, we removed the reference to province FEs from the tables. Province fixed effects now only appear in one table in the appendix and clearly states “fixed effects.” Thanks for spotting all this.

22. *There were a lot of tests in this manuscript but no mention of corrections for multiple tests. It would be helpful for the authors to mention which of these effects survive correcting for multiple tests.*

REPLY: As suggested, we have applied a correction for multiple hypothesis testing to the main results table, Table 1. Please note that the tests are not independent tests of the same theory and therefore Bonferroni-like corrections need to be performed with this in mind (Perneger, 1998). That is,

- we identified one theory-driven hypothesis in regressions 1a, 2a, 3a, and 4a, (i.e., whether participation day has an effect on the DV);
- in regressions 1b, 2b, 3b, and 4b, we identified two hypotheses (i.e., whether the interaction of participation day with either with age or gender is significant);
- in regressions 1c, 2c, 3c, and 4c, there is only one hypothesis (i.e., whether the three-way interaction *day x age x gender* is significant because the rest of the hypotheses have been tested in the previous regressions).

As a result, the only regressions that need to be corrected for having more than one hypothesis are regressions 1b, 2b, 3b, and 4b. For these regressions, the p-values are multiplied by 2. The only result affected by such multiple hypothesis testing is the estimated interaction between day and age that becomes marginally significant on donations (i.e., the corresponding p increases from 0.048 to $p=0.096$). Since there are no other relevant changes on the main results after correcting for multiple hypothesis testing, we keep the old version of Table 1 (that does not correct for multiple comparison) in the main text and only mention the corrections and the above change in a footnote.

23. *It is unclear why the authors begin referring to solidarity and envy as “SR-solidarity” and “SR-envy” as all of their data is self-report (even the donations). There are no objective measures in this study simply because the experiment was conducted online while people were stuck at home. In addition, the authors have some rationale in the introduction as to why they thought solidarity and envy would be important, but it is still not clear why these measures and not others were used.*

REPLY: You are obviously right that both are self-reported in the conventional (common wisdom) meaning of the word. However, the literature that we target distinguishes between “elicited behaviors” and “self-reported behaviors” measures. The former are typically implemented and therefore affect the wellbeing/payoffs of the subjects (i.e. our subjects really donated the stated quantity upon winning the lottery), while self-reported variables are just stated but not implemented, without really affecting the subject (i.e. our subjects state how much they care about others but the answer has not consequences for them). The main difference is that the former is considered “more objective” than the latter, which the literature terms self-reported. We just adapt this terminology to make sure that readers always bear in mind that solidarity and envy were survey self-reports (i.e. stated rather than revealed preference measures). We preserved this notation/terminology in the current version; if you insist in changing it, please, let us know.

As for the rationale for solidarity/envy, as mentioned in comment #6, the reason is very simple: we use these two for their popularity in the literature and the easiness with which they can be elicited. There is no other reason; we just made a choice. Note that any manifestation of social preferences/social concerns would present (dis)advantages and be potentially subject to criticism.

24. *The authors state “The moderation effect of age on giving is robust to controlling for self-reported solidarity and envy though.” but it is not clear what analysis the authors are referring to as Table 1 suggests envy and solidarity were in separate models and outcome variables.*

REPLY: We apologize for not being clear. This part of analyzes refers to Table A2 in the Appendix that contains all this analysis. The corresponding paragraph states: “*Finally, we use self-reported solidarity and envy as explanatory variables for giving in models 2a-2c in Table A2 (Appendix). As mentioned, solidarity and envy are significant predictors of donations in our experiment: those who self-report higher solidarity and lower envy donate more ($p < 0.001$). The moderation effect of age on giving is robust to controlling for self-reported solidarity and envy though. Model 2a indicates that the effect of participation day remains negative and statistically significant and model 2b reveals that the negative effect of the interaction dayage*

remains negative and marginally significant." We hope that this makes it clear that all these results refer to a table in the Appendix and the corresponding models within.

25. *I believe that the authors mean that exposure to public information (e.g., deaths, cases, ICU patients) causes donations to decrease. Still, the wording "donations scale down" implies that the donations decrease in a scaleable form as more access/ exposure to information is included in the model. As that is not how table 2 presents the data, more clarity is needed here.*

REPLY: To avoid any confusion, we replaced "scale down" by "decreased".

26. *It is unclear what the authors mean with "Shachat et al. (2020) perform a longer time horizon analysis in Wuhan, China and find long-term increases in prosociality after a decline in the immediate aftermath of the city lockdown. Our findings are consistent with such a pattern and, together, suggest that the behavioral adaptation process may follow complex dynamics." pg 20. The "behavioral adaptation process" aspect is not clear - do you mean changes in prosociality?*

REPLY: We rephrased the corresponding part and hope that it is clearer now. However, "behavioral adaptation process" is meant more generally here, speculating on basis of the literature that the adaptation processes—independently of whether they affect prosociality or other behaviors—might be non-linear and non-monotonic.

27. *This section is also not clear (pg. 20) "The results would also have broader implications for the building of social resilience to future disasters, for which large-scale collective response is as important as for the Covid-19 pandemic. Such challenges are numerous (Boyd et al., 2018)."*

REPLY: We rephrased the corresponding statement.

28. *By this point in the discussion, the effects of age are referred to as moderating effects - but this language was not used in the results section. It would be helpful to call the age effects similar things across all sections of the paper (for consistency from the reader's perspective, I had to go back and make sure the effect was a moderation effect instead of something else).*

REPLY: Thanks, we have fixed this.

29. *Toplak et al. is missing from the reference section.*

REPLY: Fixed. Thank you.

30. *I'm not sure if I missed it in my download, but I did not have access to the Appendix. I also was not able to access the link to the anonymous data file from the link provided.*

REPLY: We apologize for that. However, we submitted the manuscript together with the Appendix right below the reference list following the journal's guidelines. We hope that you get the new version together with the Appendix. The same holds for the data: we submitted the link to the Dryad (https://datadryad.org/stash/share/5v3SCC_FaC22DxFkCxPCy4AzRrtVlaxFHFqeNnU5Yyc) and it seems to work perfectly. If you have any problem downloading the Appendix or the data again, please, contact us directly and we will be happy to provide it to you.